# The Roles of Androgens in Humans: Biology, Metabolic Regulation and Health

**DOI:** 10.3390/ijms231911952

**Published:** 2022-10-08

**Authors:** Marià Alemany

**Affiliations:** 1Facultat de Biologia, Universitat de Barcelona, Av. Diagonal, 635, 08028 Barcelona, Catalonia, Spain; malemany@ub.edu; 2Institut de Biomedicina, Universitat de Barcelona, 08028 Barcelona, Catalonia, Spain

**Keywords:** androgens, testosterone, dehydroepiandrosterone, estradiol, dihydrotestosterone, anabolic steroids, metabolic regulation, senescence, metabolic syndrome, testosterone replacement therapy

## Abstract

Androgens are an important and diverse group of steroid hormone molecular species. They play varied functional roles, such as the control of metabolic energy fate and partition, the maintenance of skeletal and body protein and integrity and the development of brain capabilities and behavioral setup (including those factors defining maleness). In addition, androgens are the precursors of estrogens, with which they share an extensive control of the reproductive mechanisms (in both sexes). In this review, the types of androgens, their functions and signaling are tabulated and described, including some less-known functions. The close interrelationship between corticosteroids and androgens is also analyzed, centered in the adrenal cortex, together with the main feedback control systems of the hypothalamic–hypophysis–gonads axis, and its modulation by the metabolic environment, sex, age and health. Testosterone (T) is singled out because of its high synthesis rate and turnover, but also because age-related hypogonadism is a key signal for the biologically planned early obsolescence of men, and the delayed onset of a faster rate of functional losses in women after menopause. The close collaboration of T with estradiol (E2) active in the maintenance of body metabolic systems is also presented Their parallel insufficiency has been directly related to the ravages of senescence and the metabolic syndrome constellation of disorders. The clinical use of T to correct hypoandrogenism helps maintain the functionality of core metabolism, limiting excess fat deposition, sarcopenia and cognoscitive frailty (part of these effects are due to the E2 generated from T). The effectiveness of using lipophilic T esters for T replacement treatments is analyzed in depth, and the main problems derived from their application are discussed.

## 1. Introduction

In recent decades, as a consequence of the methodological advances in analysis and clinical characterization of steroid hormones, we have exponentially advanced in the understanding of their diverse functions in the control of human metabolism and behavior. Our knowledge of steroid hormone physiological functions, however, lags behind. This is due, in part to the need to combine the (predominant) pharmacologically oriented objectives, with a slower pace of advances in adequate analytical technologies. The growing complexity of the newly acquired (albeit partial and often piecemeal) knowledge is further compounded by the extension of the studies to a widening number of branching and specialized fields, such as cell/tissue compartmentation [1] and molecular biophysics [2]; this may complicate the interpretation of the metabolic-regulative picture as a whole, from fragmented (and often isolated) information.

The additional presence of bias against the “sex hormones”, for different reasons (basically unsupported by scientific knowledge) [3], and the excessive focalization of most studies on only a limited number of natural hormones [4], has left enormous gaps of knowledge about the actual functions of the steroid hormones. In contrast, the common incorporation of many drugs derived from them or mimicking some of their actions [5] continues to rise.

In humans (and most mammals), steroid hormones are conventionally classified using the classical names of their most representative molecular forms: 21C corticosteroids (i.e., glucocorticoids (GC) and mineralocorticoids), 21C progestogens (with a pregnane structure), 18C estrogens (sharing the estrane skeleton) and 19C androgens (based on the androstane structure). At present, we also include in this extensive grouping, the 24C bile acids [6] (with a cholane structure), and the 27C hydroxyl-calciferols and vitamin D vitamers [7,8] (derived from cholesterol’s 27C-cholestane skeleton, albeit not retaining its four-ring structure). There are many more steroidal hormone types, assumedly restricted to other kingdoms and phyla, but most of them remain yet to be identified and/or characterized.

The use of names for these large groups of molecules is difficult to justify when taking into account the compounds’ physiological (and more often than not, their pharmacological) effects. The molecular species within any given group are often included within broad general descriptions that try to include all of them (e.g., estrogenic activity, androgen deficit, iatrogenic effects of GC) in a short overall definition of function and belonging. The systematic use of this convention is deeply embedded in both clinical and basic science studies, commonly assuming that the different molecules may show, perhaps, a different binding ability and variable overall effects, but nonetheless, their effects remain cohesive and directly interrelated within the context of each of these groups. This interpretation is further muddled by the often scant differences on specific pharmacological actions of a number of both natural and synthetic derivatives, which, assumedly, maintain or “improve” some of the hormone functional effects (in fact, simply noting their pharmacological actions). The classical groups of steroid hormones and derived “families” need to be actualized to incorporate the ample present-day knowledge as a whole. There is a need for more precise ways to organize and differentiate (at least) the natural steroidal hormones according to their structure and function, but (mainly) taking into account the critical importance of their mechanisms of action, functions and regulation, extended to their synthetic (or location-related) pathways [9].

Androgens are a clear example of this often forsaken (or unexpected) diversity. It is generally assumed that androgens are hormones primarily related to sex/reproduction, identified with the male-oriented physiology and psychology traits and patterns. Androgens, together with estrogens (often also including progesterone), are usually also known with the old and restrictive term of “sex hormones” [4]. Androgens, consequently, are assumed to modulate body growth and differentiation following the male blueprint; they modify brain development and the functional structure along patterns (and behavior) linked to *maleness* [10], thus, establishing an, also presumed, wide gap with estrogens (i.e., the main *female* sex hormones) which characterize (and distinguish) the complex biological standing of females in the reproductive continuity [11,12]. In a social and very general sense, these simplifications may be acceptable, but they could not be used within a precise scientific or medical context. The reasons for not using them abound; for instance, the circulating levels of some estrogens are often as high in men as they are in women; in addition, children of both genders (and women) may have similar total androgen levels than (sexually functional) adult men [13]. The question is whether the term “androgens” includes a wide number of molecular species that exert many different specific functions, some of them hardly related to reproduction, at least in a direct way. The same can be said of estrogens [9]. Thus, the indiscriminate use of compounds from each of the steroid hormone groups for the treatment of a wide (and widening) range of disorders (endocrine, metabolic or even aesthetic), could not be justified without a thorough analysis of the whole, contrasting the known effects of specific molecular species with the results expected and including the probable consequences of their eventual medical application [14,15].

## 2. Age Dependence of the Biological and Social Functions of Androgens in Males

Evidently, at the core, estrogens and androgens are clearly related to reproduction—in addition to maintaining body functions and energy homeostasis [16,17]—since all these functions are inextricable from the biological drive of species’ survival. The decrease in circulating estradiol (E2) after menopause is linked to the programmed disconnection of the ovulatory cycles. Women’s age, fitness and the onset of menopause make their ability to endure (in time) the ordeal of bearing and nurturing children biologically improbable, and thus, their reproductive functions are discontinued earlier. This planned obsolescence results in “collateral” disorders affecting many systems, since estradiol helps maintain the function of a many key metabolic processes which could no longer be fully supported [18] when its availability fails.

Nevertheless, in men, the production of viable sperm (and thus, the possibility of siring descendants) is often maintained up to an advanced age [19]. This gender difference has logical nutrient-economy reasons, since the reproductive cost in time and biological resources is obviously much easier for men than the severe metabolic strain women endure in the extended time- and nutrient-expense burden of childbearing/raising. Notwithstanding, in men, reproductive activity decreases progressively with age from their apex of physical/sexual performance in youth, which is earlier than full maturity. This is due, in part, to men’s lower biological resilience, coupled with an age-related decrease in muscular power, and especially, the larger cumulative male death toll, of which is a consequence of higher risk-taking, competence for mating and the altruistic risk of defending the group against predators. This has been compounded, along evolution/history, by the risks and energy cost of family/tribe protection tasks and social role programming. Most men do not attain an advanced age, largely because of a higher cumulative exposure to mortality than women [20]. In the context of group survival, most men are expendable (and/or needed by the community for shorter lifetime periods than females). Consequently, progressive attrition lowers their ranks, socially favoring the channeling of available food resources (and protection) to fertile women and viable children. This is a common blueprint across many species, which we humans share.

Thus, in parallel to women, men are also subjected to important hormonal changes with aging [21]. Usually, the process of andropause [22] is less abrupt in its presentation than menopause is in women, but its results are nonetheless crippling [23]. The aging-generated changes limit even more the men’s “usefulness” for the group, which directly affects their individual survival. At present, however, the relative abundance of food (and better overall health) tends to partly ease this early culling. However, the few differences in the basic hormonal makeup of men and women result in a higher predisposition of men to develop the metabolic syndrome (MS) earlier and more severely [24,25]. The progressive decrease in testosterone (T) levels with advancing age contrasts with the maintained hormone cycle-sustained ovulation in women, that ceases in a short timespan at menopause [26]. MS is much more frequent in adult (and, especially, in aging) men than in women [27], but not in the very old. Menopause is associated with hyperandrogenism [28], caused largely by adrenal androgens [29], whilst lower T availability is a known critical factor for MS development in men [30,31]. This difference is indirect well-known proof that not all androgens’ functions are equal.

Hypogonadism is one of the most constant (and defining) characteristics of MS [31], which, in men, compounds the (programmed) hormonal fall caused by aging [32]. The progressive lowering of T levels elicits a cluster of negative consequences: altered body protein maintenance, decreased fitness for exercise/fight [33], diminished cognitive functioning [34], lower incentive and eligibility for mating, and a growing number of metabolic disorders. The latter are, in a significant part, a consequence of the parallel decrease in estrogen synthesis (and availability) [35] caused by the dwindling precursor T production [30]. Estrogens control energy partition [3], and their insufficient levels foment obesity [36]. This programmed obsolescence of males’ functionality in the wake of the progressive loss of T may be described *either* as a key consequence or as a main cause of MS.

## 3. Types of Androgens—Synthesis, Structure and Functions

Androgens are synthesized from cholesterol in the testes (and annex structures), largely in the Leydig cells [37] of men, as well as in the ovaries of women. The brain can also synthesize a number of androgens, including DHEA [38]. Other organs (i.e., skin, adipose tissue) produce several molecular species of androgens [39] from DHEA or other precursors [40].

In addition, androgens are massively synthesized in the adrenal cortex, in the outer *glomerulosa* layer—along with mineralocorticoids [41]—in the intermediate *fasciculata* with glucocorticoids [42] and in the inner *reticularis* layer with dehydroepiandrosterone (DHEA) [43]. The 11-keto-androgens (KTs), largely 11-keto-testosterone (KT), are major androgens [44] synthesized largely in the *fasciculata* and *reticularis* zones [13,45]. The testes are also an important site for KT synthesis [44].

Figure 1 shows the synthetic pathways of the steroid hormones in the adrenal cortex (i.e., in men and women, from a child to an elderly person), testes and other tissues (e.g., skin). Quantitatively, the main androgenic products of adrenal glands are DHEA—found mainly as its sulfate (DHEAS)—and KT, while the testes produce, essentially, T, and to a lower extent, dihydrotestosterone (DHT), but also secrete pheromones (also synthesized in the skin). Interconversions between some androgens—and essentially, the aromatization to estrogens [46]—have been described also in adipose tissue [47,48], the brain [49] and other sites [50].

Figure 2 shows the main structure/function types of natural androgens, which main synthetic pathways are shown in Figure 1. These different types of androgens are briefly described below. 

○DHEA dehydroepiandrosterone (DHEA): 7OH-DHEA and its esters: DHEAS and acyl-DHEA.○DHEA is formed from cholesterol via pregnelonone and 17OH-pregnelonone. Its hydrophilic sulfate ester, DHEAS, is the main steroid hormone in the human bloodstream [51]. DHEA binds both (albeit not strongly) the androgen (AR) and estrogen (ER) receptors [52]. Acyl-DHEA can be formed by plasma lipoproteins [53], via lecithin-cholesterol acyltransferase [54], and has been related to DHEA transport into tissues [55]. DHEA affects the regulation of corticosteroids [56,57], but all its functions have not been fully unraveled.○T testosterone (T).

T is the main and best-known androgen. It was isolated [58] and first synthesized in 1935 [59]. T is formed from DHEA via 3β- and 17β-hydroxylases; the intermediate precursors being either 5-androstenediol or 4-androstenedione (A4).

○AcT 17 β-acyl-testosterone esters (AcT).

In natural AcT, the acyl group is usually a C16-C18 fatty acid. AcT are found in small amounts in tissues [60] and, especially, in lipoproteins [61]. AcT are formed via acyl-CoA transferase esterification on C17 [62], and are assumedly hydrolyzed to fatty acids and T by a number of acyl-esterases [63]. Under physiological conditions, AcT are not aromatized to acyl-E2 [64].

○KTs 11-oxo-androgens, such as 11β-hydroxy-testosterone and 11-keto-testosterone (KT).

This group also includes the 11-oxo derivatives of 4-androstenedione and 5-dihydrotestosterone. KT is the main adrenal “true” androgen [65,66], but it is also produced by the testes [67]. The key enzyme in their synthesis is a 11β-hydroxylase, which also intervenes in the formation of GC, but (at least in the zebrafish) it favors the oxidation mode, which allows the conversion of 11-OH-T to produce KT. This mode is not adequate for the synthesis of active hydroxyl-corticoids from their 11-keto pairs, since a reducing (i.e., not oxidizing) reaction is needed. The function and regulation of this enzyme may possibly constitute an important node in the interrelationship between androgens and glucocorticoids in the adrenal glands [68]. KT is predominant, and has more androgenic effects than 11β-hydroxy-testosterone [42,69]. The formation of both 11-oxo-DHT derivatives seems to be of lesser entity than those of T [69,70].

○DHT 4,5-dihydro-testosterone (DHT).

DHT is the natural androgen [71] with the highest affinity for the androgen receptor [72,73] (AR, often referred to as “DHT receptor”). DHT is formed by a reduction in the ∆^4–5^ double bond, in the A ring of T, and by the 5α-reductase; there is an additional “backdoor” path of unclear quantitative importance that eludes the direct use of T in the formation of DHT [74]. DHT shows androgenic activity in the absence of T [75]; it is not aromatizable, and consequently, can maintain its function even under conditions of aromatase inhibition [76]. 

○A4 androstenedione (4-androstenedione, A4).

A4 androstenedione is a mild but important androgen [77]; it binds the AR with low affinity [73]. A4 is formed from DHEA or 17OH-progesterone through 3β hydroxylation [78]. It is the main precursor for the synthesis of T in adrenal cells, testicles and ovaries [77]; in the latter, T is produced this way for many years after menopause [79]. A4 is also oxidized to 17-keto-androstenedione [80], and plays an important role in the KTs metabolism [81].

○AP ∆^16–17^ androgenic pheromones: androstenone, androstenols.

AP ∆^16–17^-androgenic pheromones are a peculiar and little-studied group of androgens which do not seem to bind the AR [82]. Their main synthesis pathway is derived from pregnelonone via steroid-17-hydroxylase, 1-20-lyase and a peculiar ∆^16–17^-desaturase, followed by 3β-dehydrogenation and the action of 5α-reductase [83] to yield a unique group of molecules with a ∆^16–17^ double bond in the D-ring. This pathway is quite different from the canonic androgen synthesis [84]. Some sort of pheromone-like communication exists, in humans, based on these (and other) compounds, and basically acting in the direction from men to women [85,86,87].

○EA estrogenic androgens (e.g., 5-androstene-3α,17β-diol).

This is a polyphyletic group, characterized by their binding to the ER. However, DHEA and some of its derivatives can also bind the ER [88,89], playing an important role in the function of this ambivalent hormone. In addition, there is a small group of androgens that is structurally related to androsterone (Ane), such as the 5-androstene-diols, that bind the ER, inducing estrogenic effects [90]. It has been postulated, however, that the 5-androstene-diols (3α,17β and 3β,17β) are quite important for the androgen effects on anxiety and cognitive enhancement, which are carried out through the activation of ERβ [91,92]. The 11-Keto androstenedione—the product of the 11β–hydroxylation of Ane—also has estrogenic effects.

○Ane androsterone (5α-androsterone, Ane).

Ane is a catabolite of T and DHT, which does not bind the ER and is a weak androgen itself; however, it is a natural agonist of the FXR (farnesol X receptor), playing a role in the control of bile acid recycling and function [93], and thus, it indirectly acts on the bile acid path of energy metabolism regulation [6]. Ane has been postulated as a neurosteroid, and a mammal pheromone, but the evidence for the latter function is so far inconclusive, at least for humans [94].

○Other androgen metabolism intermediates and excretion molecular species.

This is not a specific function-directed class of androgens, but a mixed bag of intermediate or end-metabolism molecular species. A number of androgen catabolism-derived molecules have been studied as starting bases for the synthesis of (or their possible use as) anabolic drugs [95], but also as markers of androgen metabolism [96]. Most of the intermediate molecular species involved in the synthesis of androgens show mild androgenic effects, but often their possible physiological relevance as androgenic agonists has not been clarified, or even tested. On the other side, the catabolites of T and DHT have received considerable attention because of their varied metabolic effects [97,98], abundance and commercial (probably unjustified) distribution since their physiological significance has not (or perhaps not yet) been sufficiently established.

Only T, A4 and KT (albeit not DHT, but including the T eventually derived from AcT hydrolysis) are substrates for aromatase [99], i.e., they can be directly converted to estrogens: A4 → E1 (estrone); T → E2, KT → 11-keto-E2, with quite different estrogenic effects.

## 4. Mechanisms of Action of Androgens

### 4.1. Canonic Androgen Receptor Signaling

The androgen receptor (AR) is a classical nuclear receptor, primarily acting through the selective induction of the translation of DNA strands to eventually synthetize specific proteins. The AR elicits the expression of a number of genes depending on the target cells and the modulation of its signal. In any case, as it corresponds to steroid hormones, the effects of this stimulation are not immediate, but are relatively delayed in time [9]. The AR belongs to a subfamily of steroid hormone receptors closely related to the progesterone receptor, GC and aldosterone receptors (3-ketosteroid receptors) [100], and to a lesser structural but more dynamic relationship with the ER [101]. The AR gene, *NR3C4,* is in the X chromosome, and contains eight exons, which translate to proteins; the AR acts as a homodimer structure [102,103]. Two main isoforms, A (187 kDa) and B (110 kDa), have been described [104].

The AR monomer has a lineal structure which contains a small zone of high flexibility, or a hinge (Figure 3), and joins two larger arms; the longer one contains the N-terminal and the DNA-binding domains, and the shorter incorporates the C-terminal domain on the other side of the hinge [105]. The N-terminal section contains poly-Gln and poly-Gly sequences which allow for an additional, highly variable structural modification [106]. Functionally, this section contains the AF1 (activation function 1) active site [107], which binds a number of agonists, affecting the AR function depending on its nature and binding patterns, thus, extending the possibilities of regulative modulation [108,109]. The cysteine-rich DNA-binding domain is where the AR binds to the ARE (selective Androgen Response Elements) sequence of the promoter [110]. In the nucleus, the dimeric AR binds consecutive (duplicate) ARE binding sites, forming a stable AR–ARE (promoter) complex, which then bind to specific DNA sequences [111]. Other enhancer or regulatory proteins (or miRNAs) may bind to this complex of the DNA fiber, further increasing the possibilities of AR regulation [112,113]. Then, the active AR system elicits the translation of tissue-specific genes [114,115]. 

The DNA-binding domain also contains a short amino acid sequence (nuclear localization signal) which is essential for the recognition by the nuclear membrane, allowing the transfer to the nucleus of the activated AR [116,117]. The hinge domain is also susceptible of modulation by methylation, acetylation or other regulatory processes [118]. 

In the cytoplasm, the inactive AR molecules are linked to heat-shock protein chaperones [119]. When a suitable ligand binds the AR, these proteins separate from the complex. Then, the ligand-modified AR enters the nucleus crossing the nuclear membrane. In the nucleus, the AR dimer binds the promoter (ARE) and, together with other modulators, finally binds the DNA to express AR-related genes [120]. The C-terminal ligand domain contains the AF2 (activation function 2) active site in a more lipophilic environment than the AF1; this is the binding site of hormone agonists and other ligands [107]. The AF2 contains a particular depression (*ligand-binding pocket*) which binds the receptor agonists through its fitting anchorage to several specific points [121].

It has been postulated that the main agonist of the AR is DHT [122], and the AR has often been assumed to be essentially a DHT receptor, at least in prostate [123]. The higher (several-fold) binding affinity of DHT for the AR than that of T, A4 and KT [73,100] reinforces this assumption. The high prostatic activity of 5α-reductase suggests a rapid conversion of cytoplasmic T into DHA, which may increase the overall effectiveness of AR signaling, thus, resulting in a stronger response of the translation process [123,124]. However, the multiplicity of agonists, the marked difference—in terms of DHT function—between males and females, and the notable differences between tissues regarding the presence of 5α-reductase [125] and functional ARs [126,127], suggest that this process, described for prostate, could not be fully applicable to the response to T of all androgen-sensitive cells. This is just another example of the extensive ability to modulate the responses to androgens by different organs and tissues.

The AR can be further modulated by the modification of its structure, largely on the AF1 branch, by different agents, which bind or break out portions of domains (such as the SARD, or specific AR degraders) [128]. However, most of the specific binding and modulation of the AR actions are related to the N-terminal arm, which contains diverse binding sites [129,130].

Table 1 presents some of the best-known different functions of androgen classes by showing their effects on concrete specific paths or functions. Thus, physiological T and DHT effects are more marked in men than in women, probably because of their higher production in males, despite also having faster metabolic clearance rates [131]. Obviously, most androgens bind to the AR, but it is unclear whether the AcT can bind the AR on the main specific agonist site; DHEA also binds the ER [52], and the AP binds neither of these receptors [82]. However, the strength of androgens binding to SHBG is maximal for DHT: DHT >> T >> A4 > DHEA [132] (but not DHEAS) [133]. The Sertoli cells synthesize an intra-testicular T binding protein (ABP: androgen-binding protein) [134], a remnant of other less-evolved mammalian T carriers).

### 4.2. Main Non-Canonic Receptor Signaling (AcT, SARM)

The high occurrence of prostate cancer and the implication of the AR signaling in its growth [124,138] resulted in the need to find drugs able to sustain most of the anabolic and protein-protective functions of T because of the limited (or counterproductive) benefits of T deprivation [139,140,141,142]. These efforts resulted in the development of a large number of selective AR modulators (SARM) [143,144]. Most of them bind (albeit in a non-canonical way [145]) the AR (mainly at the AF2), on or close to the ligand-binding pocket [146]. Ideally, they stimulate the AR to increase muscle mass and body protein (consequently, facilitating leanness) [145,147], with limited effects on many of the other functions of T (largely sex-related) [145]. The objective of SARM design was to maintain some of the AR functions, but not completely blocking the prostate androgen-related functions (or to limit excessive prostatic activity) [148]. This mild effect was directly sought for the use of the drugs in the repression of the AR-promoted/sustained prostatic cancer [149]. The SARM can, thus, be considered anabolic drugs, but they are not necessarily steroidal, and are not aromatizable to yield estrogens [150]. The ease with which they may bind the AR *over* (i.e., not directly *on*) the core binding site, and exert a number of androgen functions, have been studied [151]; however, so far, it is unclear how the natural androgens may induce the selective modulation of the multiple actions on the AR mediated by T and DHT.

The case of AcT is more complex since they are carried in the blood in small, albeit significant, amounts [61,152], and are stored in adipose tissue [153]. The idea of converting T into a highly lipophilic waxy compound for the storage of a readily usable hormone [60,154] has been found to be partly true by the continued use of synthetic T esters as drugs for long-term androgenic treatments, since they share the structure and binding properties of the AcT described above. In any case, it is generally assumed that to induce androgenic effects, the AcT needs to be first hydrolyzed by esterases to release T. The tight spatial constrictions of the AR AF2 ligand-binding pocket depression on the AR surface limit the possibility of direct binding to this site of the large hydrophobic AcT ester molecules (at least in the way the smaller DHT and T bind) [121].

Alternatively, and following the parallelism with estrogens, acyl-estrogens show marked and distinct effects on metabolism [9,155]. Acyl-E2 are powerful estrogens as such, but acyl-E1 are not [9]. Acyl-E1, however, bind the ER [156], but not on the main ligand site, acting probably as SERM [9], and eliciting a powerful mobilization of stored or dietary lipids [157,158]. These natural estrogen acyl-esters are also found in small but significant amounts in plasma and tissues [159]. The levels of AcT in blood seem to be even lower than those of the estrogenic esters, and are found specifically in the brain [160], testes and adipose tissue [153]. Castration slowly reduces their levels in testicles and adipose tissue to the (already low) levels found in adult females [153]. At least in the brain, the AcT are synthetized by a specific 3β-5-androsterone-hydroxysteroid acyl-transferase [161,162], the main fatty acid used being stearic acid, with lauric acid following in a lower proportion [163]; both natural AcT are seldom—if ever—used in medicine. The sex differences, the specificity of the acyl moiety and that of the acyl-transferase point towards a function related with brain sex-related organization, but unfortunately, we do not have systematic studies on the metabolic effects of natural AcT. A number of short and medium-chain esters of T, as well as modified larger and cyclic acids, are used for prolonged androgenic substitution treatments. The enormously higher doses of AcT used in comparison to the natural AcT do not allow for viable comparisons, but suggest that they may act, at least, along some of the characteristic functions of androgens (i.e., those observed at pharmacological levels) [153]. These actions do not include binding the AR, since their previous hydrolysis to T is required both for oral [164] or injected T esters [165]. However, the different effects on body composition and metabolic environment caused by prolonged treatment of humans with different AcT at pharmacological levels (see Section 7.3) suggest that this is, yet, a dark zone of our knowledge, affecting, precisely, the androgens most widely used in clinical practice.

The possibility that natural, larger androgen-derived molecules could exert the effects elicited by SARMs cannot be ruled out. The parallelism of the action of SERMs and E2 with respect to the ER [9] and that of SARMs on the AR hints to the few and insufficiently studied natural AcT as possible drug candidates, given the relative differences observed between the treatment of low T availability using synthetic AcT as compared to T alone.

### 4.3. Cell Membrane-Related Androgen Signaling

In line with what can be observed in a number of steroid hormones [166,167], some effects of androgens are too fast to be justified solely by the canonic nuclear hormone mechanism of action described above for the AR [168]. In any case, at present, the most accepted proof of androgenicity is the direct binding of a given molecular species to the AR [169]. This specific aspect of most androgens is by no means an established prerequisite for androgenic signaling, despite their induction of quite a number of physiological responses. As shown below, the non-AR androgenic signaling (usually “rapid” metabolic responses) is widely extended and may be responsible for a wider and deeper spectrum of androgenic action (albeit often complementary) than the canonical path of gene expression control initiated by cytosolic AR activation. There are no data enough to quantify this non-canonic contribution, but the extensive number of studies and results hint to a well-settled series of effects that complement the other “fast” peculiarity of T and its unusually rapid turnover.

There are reports of a close association of AR to plasma membranes [170] as well as to subcellular structures, such as mitochondria [171], a situation somewhat comparable to that observed in estrogens [172]. The nature of membrane androgen signaling is essentially different from that of nuclear receptors [173]. A considerable number of studies have addressed this issue, with a higher focus on four postulated mechanisms: *SHBG-mediated signaling.* In this case, SHBG is purportedly used as an anchor for membrane AR androgen binding [174]. SHBG is produced in a number of cell types, containing its receptors in the cell surface [175,176]; they can bind the AR agonists, primarily T [175], but can also bind E2 [177]. After the agonist is bound, the SHBG-receptor-agonist complex is internalized, and then activates a membrane-related adenylate kinase, inducing the production of cAMP [176,178]. A number of effects of androgens have been attributed to these mechanisms [179,180].*Attachment to membrane G-proteins* This mechanism is based on the attachment of the AR to membrane G-proteins [170], in a way similar to that of estrogens. The binding of DHT or other agonists to AR in the membrane immediacy may then induce the activation of cytoplasmic protein kinase C, resulting in rapid effects of androgens [181]; in this case the action is also mediated by the AR, albeit not in the nucleus (i.e., not through gene expression).*Implication of ion channels*. This possibility relies on the formation of a complex of membrane G-proteins plus the AR and an agonist (as described in the previous alternative); however, this time, this affects the membrane zinc transporters [182] and the calcium channels, increasing the Ca^2+^ input [183,184]. The rise in calcium can activate the A- or MAP-kinases, inducing fast (and short-term) effects [185,186]. However, MAP-kinase and Ca^2+^ are also known modulators of gene expression, thus, AR activation may (in addition) indirectly induce genomic actions through this pathway [175].*Activation of cytoplasmic AR near the cell membrane*. A possible action on the control of membrane receptor-induced signals may result in binding AR close to the cell membrane (perhaps bound to a G-protein), previously activated by androgenic agonists and binding a tyrosine kinase [186] in healthy and tumor cells. Tyrosine kinase is also a critical modulator of the AR function [187].

This list is neither complete nor exhaustive. The simple direct binding of T/DHA to membrane-related AR has been often described [188,189]. This experimental finding, however, requires that either the signal of the AR bound to an agonist would derive to another (complementary) signal path (e.g., cAMP, Ca^2+^, phosphorylation of signaling proteins, etc.), or else it would carry the signal all the way to the nucleus. The above list shows plausible published mechanisms, but other alternatives may be found.

A clearly distinct case is that of the androgens which do not bind the AR, since none of the alternatives implying the nuclear receptor may explain their actual mechanism of action. This leaves out the DHEA (in part), AcT, AP and EA types (Figure 2). The case of EA is obvious, since their target is the ER, not the AR, and they are listed here only because of their androgenic origin (but not function) and for the absence of an aromatic A ring, the most differentiating estrogenic structural characteristic.

DHEA (and/or DHEAS) is quite peculiar, since it is basically a promoter hormone [190,191], acting through interactions with multiple hormonal receptors [52,89], with mild androgenic and estrogenic effects [89]. DHEA has a marked anti-glucocorticoid activity [56,57] that, at least in part, may be explained by its structure [192]. Its physiological effects are wide and varied; DHEA is a principal neurosteroid [193,194] which participates in the regulation of neuronal apoptosis [195] and is an antagonist of the GABA A receptor [196]. In addition to its key role in the synthesis of androgens and estrogens, it helps maintain hormonal (and substrate) homeostasis [197], and participates in a number of different functions, such as anti-inflammatory [198], stress control [199] or erythropoiesis [200]. Its high implication in human health and disease [51,201] helps set this hormone away from other mainstay androgens because of its limited *classic* androgenicity.

In parallel to T, however, DHEA can be made more hydrophobic by esterification with fatty acids in plasma lipoproteins [53], in this case via lecithin: cholesterol acyl-transferase [54]. The possible function of these low concentration steroid derivatives [54] has not been yet clarified.

Probably the most curious mechanism of androgen signaling is that of the AP. Their effects (through chemical signaling) are not exerted on the same organism producing them, but act on other (remote) individuals. The AP have been proposed as human pheromones [202,203] because of their synthesis in the skin [204], volatility and odorous signaling [87,205]. Despite being commercially sold as “sex attractants”, their known effects on other (non-ape) species’ vomeronasal organs [206] have not been fully proven in humans because the presence of this organ is—at most—vestigial [207,208]. Nevertheless, AP receptors in the nasal epithelium have been directly found to signal the close-by hypothalamus [209]. 

Androgenic pheromones are unrelated to the canonic AR [82]. In many mammals, including primates, the AP may bind protein receptors in the nasal epithelial cell surface, which genes are conserved in humans [210], such as G-proteins [211]. Binding activates ion channel systems, such as the TRP (transient receptor proteins) [212], which activate neural circuits [213]. The receptor-putative pheromone ligands are generally assumed to possibly play a role in this area [214] full of uncertain homologies with other animals, difficult investigation procedures and a limited availability of hard data. The fact that the AP modulate GABA channels [215], the ubiquitous distribution of TRP and a variety of cell AP-binding “receptors”, suggest that AP may play other regulative functions through interpersonal signaling of social or collective situations, such as the transmission of information on food (or sexual) availability [216].

## 5. The Varied Physiological Functions of Androgens

The described (direct or indirect) interactions of androgens with the normal functions of metabolism and its regulation, both in health and under altered situations, cover practically every possibility for intervention or interaction. The variety of mechanisms described in Section 4.3, and the control of gene expression through the activation of AR, often results in widely different effects. The high number of gene targets may be further modulated by AR responses to the varied different molecular species of androgens, resulting in an exponential continuum of modulatory possibilities. Despite our yet limited knowledge of these variations, there are a few general targets/mechanisms, in which it is generally accepted that androgens intervene, eliciting significant effects by action or restriction.

### 5.1. Estrogen Synthesis

The androgens are a necessary previous step for the synthesis of active estrogens in sufficient amounts, especially its main representative, E2. Consequently, the actions of estrogens derived from androgen precursors may help explain a growing number of apparently paradoxical or opposed “androgen” effects. It is important to note that T is the main substrate for estrogen synthesis, which is not a reversible process. The metabolic actions of T are in part a consequence of its conversion to E2 [217,218], and low T levels are correlated with increased insulin resistance [219,220]. However, E2 decreases insulin resistance [221] and T, whereby low levels are associated with diminished insulin sensitivity [220], and may elicit an increase in insulin resistance in the WAT of female rats [222]. T also acts on glucose metabolism via aromatization to E2 [223], as explained in depth in a previous review [9]. Furthermore, E2 also exerts critical cell-protecting anti-oxidative effects [35]. On the other side, glucocorticoids increase insulin resistance [224], favoring the deposition of body fat.

### 5.2. The Complex “Love-Hate” Interactions of Androgens and GC

The androgens and GC families of steroid hormones tend to favor the maintenance of homeostasis; however, they exert their actions using different, complementary (albeit often opposite) processes. In any case, both groups of steroids help maintain energy, substrate and global homeostasis, despite their functional constrictions, by normalizing the consequences of the variability of the nature and availability of substrates. The interaction of androgens with GC needs a further direct and deep study because of often disparate interpretations caused by the unperceived complexity of their continuous interactions.

DHEA is a known (and important) inhibitor of corticosteroid action [57], in part by interfering with the glucocorticoid receptor (GR) function [225], and counteracting the effects of GC [56]. However, the GC decreases the synthesis of T via downregulation of gonadotropins [226], thus, decreasing T-related sex responses [227] and spermatogenesis [228].

Figure 4 shows the close interrelationship between GC and androgens (and estrogens) in the adrenal glands. The key point is the sharing of a few regulatory enzymes in GC and androgen synthesis/activation pathways, thus, able to inversely enhance (or diminish) the hormonal output of androgens (and estrogens) or GC. Adrenal androgens (i.e., KT) are quantitatively important and are synthesized by adrenal glands and gonads [13,65,66]. At least in women, their main site of production are the adrenal glands [66,229,230], via 11β-hydroxylase [45,231], which also promote the specific synthesis of GC (by oxidizing C11) that characterizes this group of hormones. The interconversion of 11-hydroxy- and 11-keto-androgens is catalyzed by 11β-hydroxysteroid dehydrogenase [232]—again, the same enzyme which interconverts the hydroxyl and keto forms of GC [233], playing a critical role in the regulation of GC activity [234]. Active androgens (such as KTs vs. its less active 11β-OH-T paired molecule [232]) and inactive GC (such as. cortisone vs. its active form. cortisol) are interchanged (and their predominant function—regulated) by the same enzyme, 11β-hydroxysteroid dehydrogenase (11βHSDH), which may determine the predominance of their counteractive regulatory functions on energy partition and protein N maintenance [227], despite collaborative synergy [235] on specific situations such as tissue repair and energy homeostasis [236]. This is due, in part, to GH secretion (activated by GC [237]) and the accrual and preservation of body N by T [238].

A critical point is the cross-reverted inhibition due to the predominance of synthesis/degradation of androgens and corticosteroids in their active forms because of the common sharing of 11βHSDH. Overall tissue oxidative drive (i.e., decreased production of NADPH) favors the conversion of the 11-hydroxyl groups to 11-keto (the oxidized form, producing NADPH) of both androgens and corticosteroids. Thus, in oxidative mode, the enzyme–coenzyme system drives, in parallel, an increase in the androgen response (by enhancing KT production) and a decrease in GC activity by lowering the response through the conversion of cortisol or corticosterone to their much less active pairs (cortisone and 11-dihydro-corticosterone), and consuming NADPH to yield NADP^+^. 

A parallel situation of a single enzyme regulating in opposite directions (also shown in Figure 4) is the enhancement of androgen action; the consequent decrease in GC activity is the case of 5α-reductase [239]. This enzyme reduces T (consuming NADPH) to the more powerful DHT (thus, enhancing overall androgenicity), but also acts on cortisol (or corticosterone), reducing them to their inactive catabolites 5α-dihydro-cortisol and 5α-dihydro-corticosterone. The 5α-reductase acts on a larger number of steroid hormones (and drugs), inducing deep shifts in the regulation of the corresponding hormone action. Again, a reductive action of the enzyme, consuming NADPH (i.e., reductive mode), results in the loss of GC activity and a global increase in the ability to stimulate the AR, since DHT has much more affinity for the AR than T.

There are some similarities between the functions of androgens and GC. They are derived, in part, from their common precursor pregnelonone, and they share a fair proportion of the enzymes implied in their synthetic pathways. This likeness of sorts can be seen too in the largely shared structure of their receptors, since they belong to the 3-ketosteroid receptors (3-C) group (which includes the AR, GR, mineralocorticoid receptor and progesterone receptor) [100]. This common structure of receptors is not shared by the ER, or estrogen-related receptors (3-B) group comprising only the ERα, β and γ types [100]. Despite all being nuclear receptors, the structural genetic and regulative differences are not the same, and are largely marked by their physiological agonists. In this case, the main differencing factor may be found in the irreversible phenolic nature of the estrogen A-ring, necessarily affecting the shape of the receptor’s binding site.

The CG are related to the appearance and development of depression [240,241], and androgens are known to lower the severity of this disorder [217]. GC are secreted as a response to stress [240], in which body fat and glucose availability are relatively protected [242,243] at the expense of growth and body protein [244], with the maintenance sustained by androgens [238,245]. Nevertheless, the metabolic effects of KT (and T) share (at least in part) the homeostatic growth-promoting effects of GC, insulin and GH [236]. The protein synthesis-linked anabolic nature of T, KT and DHT [245] contrasts with the drive of GC to favor protein scraping to fuel glucose availability [246,247].

### 5.3. Modulation of the Immune Response

There is an important implication of androgens in the shaping of the—sex-dependent—immune response [248], which also affects T [249]. Most of these interactions of T inhibit the immune response [250]. However, both androgens and GC protect and enhance specific facets of the immune function [251,252], acting through different paths to enhance protection, despite their mode of action not being coincident but complementary [253,254]. The DHEA inhibition of overall GC action is a part of these finely balanced mechanisms that prevent excess defensive actions in situations where they may be counterproductive.

### 5.4. Androgens and Reproduction

The implication of androgens in the reproductive process remains, probably, their best-known global function. However, their actions can be further dissected (despite considerable overlapping) in:**The production of spermatozoa**. Androgens elicit the production and maturation of spermatozoa through the activation of the testicle Sertoli cells [255,256,257] and their incorporation to the seminal fluid. The use of large doses (and extended treatments) of T, AcT or DHT (but also E2) partially block the action of FSH [255]. Both FSH and testosterone are needed to produce viable spermatozoa [255,258]; the alteration of this delicate process often destabilizes spermatogenesis, resulting in sterility [259].**The development of male sexual secondary characteristics**. The male secondary sex characteristics are typically induced by T [230], but are generally attributed to DHT [260,261] (rather than T, which obviously also induces them [262]), because of the higher response elicited by DHT, which seem final for this purpose in comparison with the T intervention in a wide panoply of functions and paths. T seems to stimulate the defining characteristics of “maleness”, at least in part, via its conversion by 5α-reductase to DHT [263], where isozymes are considered to be responsible for most of the male-defining physical and behavioral characteristics [264]. The genetic absence of 5α-reductase isoforms may even result in individuals arriving to adolescence with a full female phenotype. However, in many cases, they become fully fertile men with male behavioral orientation, albeit they are initially devoid of secondary male sex characteristics; they often recover/develop the necessary functional and anatomical structures to allow successful mating [265] and impregnation. Nevertheless, sexual development is probably induced by a full collaborative effort of most types of androgens (except AP, AE, and probably natural AcT), and requires the presence of E2 in addition to T and/or DHT [266]; this conjoint action includes successful spermatogenesis [267].**Interpersonal communication and signaling**. This is a process in which intervene the AP, sex-specific pheromone [85,87]. Some of these compounds display neurosteroid functions [215], but they mostly have been related to inter-individual communication as pheromones [87,202,268] and/or odorous markers [205]. They are produced by and secreted from skin apocrine glands and other organs such as the brain, adrenals, ovaries and testes [83,215], and have been detected in circulating boar plasma [269].**The focussing of brain structure/function (and behavior) towards reproduction**. This includes extensive behavioral and social energy (and time) investments [270,271]. T is necessary for mating and the maintenance of a sexual/affective relationship for both sexes [272]. T activates mating relationships [273] through complex mechanisms in which cortisol [274] and reward systems, such as endorphins [275] and oxytocin [276], play important roles. The settling of durable relationships result in a decrease in T [277] and increase in oxytocin [278,279] levels in males. This change may be, in part, justified by the need to temper the aggressiveness and dominance drive elicited by high testosterone [280] or DHT. The varied effects of DHEA [281], T and DHT on sexual drive are well known, but DHEA effects are more extensive in females than in males [282], and those of DHT are practically circumscribed to men due to the direct relationship of DHT with male secondary sex characteristics. In women, DHT levels are normally low [283], rising only in some pathologies [284] and provoking serious metabolic complications in animal models when administered (i.e., PCOS [285]). Consequently, the activation and maintenance of libido is largely centered on T, for both men [286] and women [287]. The implication of KTs in these sex-related processes has not yet been clarified, but we can hypothesize that they may not be critical on this issue, since the highest circulating levels of KTs are found in childhood (in both sexes), i.e., before sexual maturity. In any case, the effects of T on the libido of women correlate with T administration/levels [288] (as in males), irrespective of their already higher and maintained KT levels [289].**The potentiation of growth and development**. This is achieved through gender-related differentiation; this is especially marked on the effects induced on brain organization [290,291], behavior [270,292] and fulfillment of the biological male phenotype [293]. The effects may be quite different with DHEA, which neurosteroid nature [194] and diverse brain effects range from behavior (i.e., aggression [294] or mood [281]) to cognition [281,295]. There is scant information on the possible effects of DHT on the nervous system, but in the brain, T reduction to DHT has been described [296]. Similarly, the known role of KTs on fish neural development has not yet been observed (albeit it is hinted to [297]) in humans.**The accrual, maintenance and regulation of body (i.e., muscle) protein**. Androgens play a critical participation in the regulation of body protein content (including, especially, muscle mass and distribution) [245]. The marked decay of testosterone availability with senescence [298] lowers muscle mass and function [299] down to sarcopenia [300]. The consequences are compounded by the limitation of estrogen production because of insufficient T. The use of T as senolytic helps limit the consequences of this deficiency [301,302]. Androgens are commonly used as drugs for the development, growth and maintenance of body protein, especially muscle mass [245,303]. This is a complex process in which other hormones intervene, such as insulin [304], growth hormones [305] and estrogens [306], and is dependent on the adequate supply of dietary energy and protein. The main androgen agents favoring body protein accrual and maintenance are KTs, T and DHT; the intervention of AcT is also probable because of their unique natural long half-life, but no specific information is available on their effects under physiological conditions. However, there is considerable evidence of the protein sparing/mass-enhancing effects of long-term TRT (T replacement therapy) [307,308], including their use for the treatment of sarcopenia [309,310]. Furthermore, the abuse of AcT as anabolic agents for sport doping, or body muscular build-up (i.e., often for non-health-related purposes) has shown that their overuse indeed results in an exacerbated growth of muscle mass [311,312] partly at the expense of body fat [313]. Their use may also result in the development of dependence [314]; often inducing severe cardiovascular, behavioral and reproductive disorders too [315].**Energy partition and handling**. Androgens are directly implicated in the mechanisms of energy partition and utilization for metabolic function. They participate in the intertwined regulation of energy metabolism with estrogens and other regulatory hormones, such as GH [305,316], insulin [317,318], calciferols [319] and cytokines such as leptin [317], but also favoring an anti-inflammatory vs. inflammatory cytokine distribution [320]. These actions have been essentially described for T and KT, but can also be elicited by AcT-based TRT [321]. Hypoandrogenism is correlated with obesity [322] and it is one of the key MS disorders (and markers) [323]. Consequently, the “recovery” of androgen levels (lost to age and/or MS) may be expected to favor the shedding of excess body fat. In fact, treatments using T decrease adiposity [324,325,326]. However, longer treatments with exogenous AcT may reduce body fat, but not massively [327,328,329,330]. Some estrogens (such as E2 and acyl-E1) are known to lower body fat [9], and hypoandrogenism results in the insufficient availability/circulation of E2 [331] because the lack of T deprives the process of aromatization to E2 of its main substrate. It can be assumed that the “adipolytic” effect of T is probably (or, potentially, mainly) a consequence of the restitution to normalcy of E2 levels [9] elicited by the T administration. Thus, only aromatizable androgens may be expected to significantly influence body fat when used for substitutive androgenization treatments. A low T is correlated with lower insulin sensitivity in men [332,333]; insulin resistance does not affect T but the reverse is true [219], since T lowers insulin resistance [220], and helps maintain glycaemia [334,335]. DHEA also decrease insulin resistance [336,337]. However, DHT (in men) has been found to increase insulin sensitivity [317,338]. The effects on insulin sensitivity/resistance induced by long-term pharmacological treatment with T and AcT are presented and discussed in Section 7.2. Effects on insulin resistance/sensitivity were observed both using T- [325,335,339] and AcT-based [329,330,340] TRTs.

This list, evidently, shows the existence of a large number of redundant, or even contradictory actions, but when compared with the chemical-functional groups described in Figure 2, a clear trend can be established between molecular structures and the general functions of the different androgen classes.

## 6. Regulation of Testosterone Synthesis and Availability

### 6.1. Hypothalamus–Hypophysis Axis Regulation of Androgens

In humans and other primates, the main androgens (i.e., DHT, T and KT) and estrogen (E2) are partially transported in blood by SHBG [133,341] instead of ABP (regulation not shown in Figure 5), but also by albumin and other proteins [342]. SHBG is a complex protein, with multiple functional forms, whereby the affinity for hormones is regulated in part via proteolysis [343], or the binding of non-steroidal ligands [344]. The transporting role is complemented by its function in signal transduction, or as a marker-transductor of other metabolic-regulation functions such as those described in Section 4.3 in relation to androgen signaling.

The circulating levels of T, E2, KT and DHT modulate the pulse secretion of GnRH (gonadotropin-releasing hormone) from the hypothalamus [345,346,347] directly into the hypophysis portal vein system [348], despite GnRH regulatory effects extending to other organs of the HHG (hypothalamus–hypophysis–gonads) axis [349]. The hypothalamus also releases the antithetic hormone GnIH (gonadotropin-inhibitory hormone), which inhibits the secretion of hypothalamic gonadotropins [350]. Prolactin also inhibits GnRH action by lowering gonadotropin secretion [351]. 

An even more complex and physiologically important mechanism to control the release in gonadotropins (in addition to direct neural/neurochemical signaling) is the effect of glucocorticoids, such as cortisol as blockers of the hypophysis response to GnRH [352] in part through the regulation of its gene expression [353]. This effect, complemented by the direct action of glucocorticoids inhibiting testicular functions [354], completes the link between stress and metabolic strife with the modulation of androgen–estrogen availability and the reproductive function. 

This strong controlling relationship between glucocorticoids and androgens/estrogens is critically centered in the adrenal cortex (as explained in Section 5.2), thus, the shared (but opposite) adrenal control of corticosteroids and androgens also has a brain connection, or an additional control node, in the hypothalamus. CRH (corticotropin-releasing hormone) is secreted from the hypothalamus following the direct portal vein path to the hypophysis, but along an even more complex circuit than that of GnRH [355]. CRH elicits the release of ACTH (corticotropin or adrenocorticotropic hormone) by the hypophysis. The process is controlled by a feedback loop based on blood glucocorticoid levels [356], parallel to that described for T and LH. ACTH is the main hormone controlling the synthesis of corticosteroids in the adrenal glands, but we lack information on whether ACTH also regulates the adrenal production of androgens (and consequently, of estrogens).

Figure 5 presents a general view of the main system controlling the circulating levels of T (and in general, of the principal androgens). T is mainly synthesized by the Leydig cells of the testes [357], but also by the prostate, epididymis, ovary and adrenal glands [358,359,360]. DHT is produced largely (albeit not exclusively) in the testes and annex structures, and the KT in the testes and adrenal glands [44]. Part of the T is transported, within the testes, by a specific T-carrier protein, ABP (androgen-binding protein) [361]. This protein, excreted by the Sertoli cells [362], is found in humans almost exclusively in the testes (it is also expressed in cardiomyocytes [363]), but is a common (main, because of the absence—or almost—of SHBG [364]) steroid hormone-transporting protein in many other animal species [364,365]; the family of ABP-transporters is subjected to a considerable evolutionary pressure [365]. ABP binds T with high affinity [366], and helps maintain the availability of T to allow spermatogenesis in the Sertoli cells [367,368] even under conditions of low T availability.

The posterior lobe of hypophysis responds to the GnRH stimuli by increasing the production (and release to systemic blood) of gonadotropins: LH (luteotropic hormone) and FSH (follicle-stimulating hormone). These hormones play a key role in the regulation of the ovarian cycle in females, but also control, respectively, the testicular synthesis of androgens in Leydig cells [357] and spermatogenesis by the Sertoli cells [258], the latter with the necessary intervention of T [367] and E2 [267]. LH secretion is the direct mechanism for the activation of gonadal steroidogenesis, modulated by progesterone [369] and cortisol [352], but mainly through essential direct feedback regulation by T, E2 and DHT [370]. Spermatogenesis is a more complex issue, and the factors which regulate the process are multiple and intertwined [371,372], including local additional regulative processes in the testes [373].

### 6.2. Hormone Availability, Interactions and Turnover

The control of T levels (and those of DHT, and largely, E2) is dependent on the balance between the synthesis of T and its (hepatic-intestinal) inactivation [374,375]. However, it also depends on its irreversible conversion to other androgens (DHT, KT) or aromatization to E2, since none of these processes could compensate the loss of T. Thus, in an adult male, the turnover of (mainly) testes-derived T needs to be fairly high [376], with a T production in the range of mg/day. This large amount of T is needed to accomplish most of the functions described above; consequently, the incapacity to meet these needs results in hypoandrogenism/hypogonadism [377]. The insufficient supply of T (and E2) affects Sertoli cell spermatogenesis [378], eliciting a lower mass and functionality of testes [379,380]. However, this also results in negative feedback on the Leydig cells, producing even less androgen, and consequently, aggravating the hypoandrogenic situation. The initial cause of low T availability may be either a failure of the testicle to produce T—because of a hormonal, developmental or toxic disorder. But it may be due to an insufficient response/production of gonadotropins, mainly LH [381], or a lower effectiveness of LH, enough to stimulate the synthesis and release of T [382]). It is possible, too, that the progressive decrease in T along maturity and old age in men [383,384], aggravated in MS [385,386,387], may contribute significantly to this snowballing of metabolic disorders caused by the loss of available T.

Substitutive treatment of hypotestosteronemia may help improve the situation by adding an exogenous source of T (if what is given is indeed T) to limit or reverse the fall in circulating T. However, in biology, things seldom are so simple and direct. Theoretically, giving T as a drug helps sustain its circulating levels, but its maintenance along time may make redundant the natural HHG regulation of its synthesis, affecting, this way, the critical function of producing spermatozoa for reproduction [388]. The pharmacologically increased T will probably inhibit the secretion of GnRH and gonadotropins, at least on a system in which the low T levels should (and could) activate the eventual higher production of GnRH—and then LH—to enhance the natural synthesis of T. A relatively short androgenic drug treatment may allow for the ulterior recovery of testicular function [389], but excessively long-term treatments may induce more severe and enduring damage to the HHG axis regulation, affecting fertility and even inducing sterility [390,391]. Thus, it is critical to keep in mind the need to reach equilibrium between preserving regulatory and testicular functions and the need to provide exogenous T to fuel them, including the “recovery” of the full operability of the testes themselves. The administration of T for relatively short periods should result in the maintenance of circulating T (also for short periods), but the partially preserved feedback loop could hopefully induce a decrease in gonadotropin secretion, thus, further limiting the falsely “unnecessary” (because of the feedback inhibitory signals of the exogenous T) internal production of T. Nevertheless, the inclusion of regular HHG-activating “interludes with no-exogenous-T” along the treatment may force the HHG axis to step in to cover for the withdrawal of exogenous T. This way, the system should be forced (if operative) to reactivate gonadotropin stimulation. This reset induced by the “rest” periods of treatment has two theoretical advantages: a) reinstate the natural regulation of T levels to a body being adapted to higher T availability and b) to prevent further (possibly long-lasting) damage to the whole regulatory system. Despite the endogenous T production being probably insufficient, at least the regulatory mechanisms may be maintained, and a new period of exogenous T may help to increasingly restore the so far poorly active endogenous T synthesis and regulation system. The possibilities of the feed-back procedure to restore the functional HHG axis regulation are, in any case, better than those of an uninterrupted full-extended long pharmacological TRT. This approach has the advantage of maintaining spermatogenesis due to the high affinity of ABP for T, which seems to help spermatogenesis to function even under fairly low circulating levels of T [392].

There is probably a parallelism between the male hypoandrogenism of maturity and the female “hyperandrogenism” conditions of POCS (polycystic ovarian syndrome) and post-menopause. In women, the critical hormone classically insufficiently affects E2 mainly, a main factor responsible of the efficient control of energy metabolism and reproduction. There are at least four important converging processes justifying the E2 deficit: (a) the lower relative availability of T, especially in the ovary after menopause [393,394]; (b) insufficient channeling of A4 to T (and then to E2) in the adrenal glands of POCS patients [395], showing higher A4 and E1 levels in the glands and plasma [396]; (c) decreased aromatase function induced (in POCS) by FSH in adrenal gland *granulosa* cells [397]; (d) acute interference of glucocorticoids over FSH in *granulosa* cells affecting aromatase [398]. The combined result is a global decrease in aromatase activity, and the predominance of A4 over T as key androgen in adrenal glands (and in part in the blood). In some way, compounding: (a) the limitation of DHEA availability, (b) the probable setting of adrenal glands in a “glucocorticoid” mode (high CG/low androgen) and (c) the limited capability of the ovary to produce enough T. In these two quite different metabolic situations, the result is a lower aromatase activity on T → E2, but is displaced to produce a relative excess of E1 from A4. The conversion of E1 to E2 is problematic, and E1 could not substitute E2 in most of its functions. One of the consequences of this adrenal disarray is the accumulation of unused A4, despite lower synthesis. The loss of endocrine ovary functions could not be substituted by the adrenal glands with respect to estrogen synthesis (more E2 is needed, not E1) in a way comparable to the lower testicular endocrine activity that could not either be fully compensated in many critical functions by the adrenal cortex.

## 7. Current Pharmacological Utilization of Androgens

### 7.1. The First Studies: T as Senolytic/Energizer

Even before the full hormonal nature and discovery of T were established, preparations (“extracts”) from animals’ testicles were used [399] as a possible treatment for the loss of sexual vigor, muscle power, clarity of mind and many other functions already attributed to youth via fully functional testes [400]. The earlier studies—including experimental self-administration of testicular preparations, with tens of thousands of elders expecting “rejuvenation” [401]—used ill-defined and aleatory “extracts” from pigs, bulls, monkeys, rats and a long list of unfortunate suppliers [402] (including human cadavers). Evidently, there were no adequate controls (and the identity of the wonder agent was unknown), but the rapid (and short-lived) “miraculous” effects were highly publicized and avidly followed [403]. Intents to rationalize the studies, and the surge of medium-term serious sequelae (immunity reactions, rapid loss of effectivity, severe complications, erratic availability of sources, social squealing, etc.), terminated this first bout of exploration. In any case, the seed of this line of work continued to grow, arriving at the purification of T [404] shortly afterwards. The next step—necessary to supply the growing demand of T—was its complete synthesis [405]. Additionally, from then on, its generalized use (despite not being universally available) started with two main applications: senolytic (anti-aging/gerontology, but also sexual enhancer) and anabolic (bodybuilding/sport-doping), which continue to be developed at present.

### 7.2. The Practical Difficulties of Oral Administration of T

The main problem that poses the direct oral use of T is its rapid first-pass inactivation by the liver [374]. Most steroid hormones (including T) are well absorbed from the intestine aqueous medium [406]. Intestinal cells and microbiota may modify the steroid molecules, as is the case of bile salts, by breaking them up, freeing the bile acids and/or modifying their molecular structure. This process may be important for the use of AcT by releasing free T (which may be further modified and/or absorbed [407]). The intestinal T is carried in part via lymph [408], but most of it is essentially transported by the blood to the liver via a portal vein. There, T is massively inactivated in the first pass [406,409], yielding oxidized derivatives [410] and/or esterified to T-glucuronate [411] since these compounds are mostly inactive and excreted [406]. Therefore, only a small part of oral T can reach (intact) the systemic circulation from the hepatic venous efflux and lymph (and thus, able to reach the brain, muscle, testes and other organs and tissues), because most of the ingested T had been previously inactivated. The interactions of T with the gut and its microbiota follow a crossed regulatory system akin to that found between the microbiota and other regulatory axes [412]. 

The T produced in one day by adult humans amounts to 7–8 mg in men, and about 0.5 mg in women [131,376,413,414,415]; this is a fairly high figure for a hormone that has a fast turnover, in the range of minutes [416]. However, its direct (mainly gonadal) secretion into the systemic bloodstream helps circumvent the hepatic portal system trap.

The adrenal cortex production of KTs again eludes the direct first-pass through the liver, but we lack sufficient information on the disposal of this important source of mainstream androgens. To limit the problems of hepatic inactivation, pharmacological procedures for the application of T were initially explored and developed [321,417]: (A) The direct deep intramuscular injection of T was dissolved in an oily excipient to create a tissue-embedded depot of T, which was expected to slowly release the hormone [418]; (B) The oral administration of T, in which the C17 hydroxyl group was esterified with organic acids (i.e., AcT) [419], and thus, was in part transported via lymph (because of the added hydrophobicity [408]), therefore, limiting the liver inactivation because the only T hydroxyl group was already “occupied” by a lipophilic acid, and thus, could not be glucuronized. Consequently, a fair number of T 17-hydroxy esters were synthesized [417]. However, some T was released by the action of intestinal esterases on these AcT, and the free T was then subjected to the same problem: hepatic first pass losses, as free oral T did [420]; (C) These same T esters were also injected into tissues dissolved in lipophilic carriers, mainly for medium- or long-term depot treatments. The length and nature of the esterifying acid affected its lipophilia and the estimated resistance to esterase action, being used to adjust the supply of T in clinical applications [59,321,417]; (D) Maintaining T as a main molecule and supplying it using repeated trans-dermal procedures, such as the use of hydro-alcoholic gels [421], skin sprays [421] or patches of varying releasing potency, efficiency and distribution [422], slow-releasing T pellets [423], silicone implants [424] or subcutaneous injections [425]. This last approach also circumvents the initial passage through the liver, since the T (or AcT) was absorbed slowly over large surfaces and not through the gut–liver system. Initial studies specifically used testicular patches [426], but the incommodity of their use made this galenic preparation fairly unpopular; (E) Buccal adhesives, oral tablets [427] and other trans-mucosal approaches [321].

Overall, many of these procedures were not well tolerated by the patients because of the long treatment periods and the uncomfortable and frequent application procedures.

### 7.3. The Widely Extended Use of T Acyl-Esters in the Treatment of Hypoandrogenism in Men

A considerable number of pharmacological preparations for the administration of T have been developed [321,417]. In addition to the use of free T, a large part of the TRTs rely in the modification of T by esterification (i.e., producing androgen drugs structurally akin to AcT) using [428] short, branched, odd-carbon, unsaturated or cyclic fatty acids. The products were designed to limit the hydrolysis of the ester bond (in fact acquiring a better control of the release timing) and were assumed to liberate T lineally over time. In addition, the esterification provides more stability to the drug, because it prevents glucuronization and limits oxidation, thus, extending the effectiveness of the “T pharmacological preparation” over time.

The main T esters used in TRT (or as anabolic drugs) are, nowadays, largely centered on T undecanoate [429,430,431], but also include propionate [432,433,434,435], isobutyrate [436], oleate—a natural AcT—[437], enanthate [438,439,440], undecenoate [441], cypionate [442], butyryl-cyclocarbonate [443], phenyl-propionate and decanoate [444]. Gradual-release T ester blends are also used to obtain extended time-uniform T release patterns, as is the case of Sustanon 250 [445,446] (a mixture of T propionate, isocaproate, phenyl-propionate and decanoate).

The direct deep i.m. injection of T or T esters (usually dissolved in edible oil—triacyl-glycerols (TAG)—or a similar, neutral, slowly dissolved hydrophobic metabolizable vehicle, is at present, probably the method most often used for the administration of androgens [447]. The intramuscular lipid blob is slowly disintegrated via phagocyte or tissue extracellular lipases/esterases, which hydrolyze the excipient acyl-glycerols and the T esters, yielding T, fatty acids and glycerol. However, its pathogenic persistence may occur [448]. The presence of acyl-steroid esterases in many tissues [63], including liver, intestine (and the microbiota), may liberate T from its orally ingested esters. A large part of oral testosterone undecanoate is hydrolyzed to T just in the intestinal wall [420], and then it is freely absorbed and distributed at least in a small significant part via lymph [408]. The enzymatic dissolution of the oily TAG deposit excipient exposes the AcT, which are then in part carried by the blood and may act as a natural AcT—this is a question that so far has not been sufficiently studied. It is generally assumed that the released AcT are rapidly hydrolyzed by tissue or cell esterases (probably including unspecific sterol-esterases). The result is a more or less steady flow of T from the injection site to the systemic circulation [449]. The injection methods have the advantage of easy administration, needing only a few applications per year to maintain a theoretically regular and fairly stable flow of androgen to the blood [449]. On the negative side, the volume of the injected preparation may induce a local focus of disturbance affecting the surrounding tissue and blood vessels. Nevertheless, the main direct problem caused by this method of administration is probably its relative irreversibility. After injection, the drug will keep being released from its depot for a long time at an expected estimated (albeit diminishing) rate, dependent on the mass of AcT remaining in the injection site, a process that could not be stopped.

The (usually) daily transdermal administration of T using patches [450,451] or the extension of hydro-alcoholic gels over bare skin [452,453] have the advantage of relative immediacy, and the possibility to control the time of exposure to the hormone [454]. Its main problems are the cumbersome procedure of extension (gel) or the limited area of application (patches). There is an added difficulty using this approach: the estimation of the dose actually received by the patient, since only a fraction of the T is in contact with the skin, is eventually absorbed and carried by the systemic circulation [455]. Furthermore, the dose may be further diminished because the skin can metabolize T to produce DHT and other androgens (such as AP); thus, repeated application (direct exposure) may result in local unwanted effects on the skin and the integumentary system [456]. In the case of gels, care must be taken for (unexpected) delayed T transfer to other individuals’ skin or mucosae through direct contact with T-loaded skin [457]. In any case, a clear advantage of the use of gels is the easy way to stop the treatment: simply washing out the hormone not yet absorbed.

A number of other approaches to an efficient, safe (and comfortable) method of administration of T have been postulated and commercialized [451]: subcutaneous pellets [423], nasal gel or other applications [458,459], buccal tablets [460], subcutaneous injections [425], new oral preparations [427], combined transdermal and injected T [461], etc. In all cases, we encounter the same inactivation barrier problems (liver, digestive tract), and/or the difficulties of assessing concrete and precise doses of T for defined periods of time. An important factor is the easiness and comfortability of the administration, and the compliance of the patients during the often very long periods of treatment (including the danger of overdosing because of the time needed to achieve the expected results).

Last, but not least, most “testosterone” drug preparations *are* indeed AcT. However, T and AcT are different androgen types, and in most cases, we do not know yet whether their physiological and pharmacological effects are equivalent. In fact, we have reasons to believe they are not, based on the data and analysis of their different effects presented in Section 7.4 and Section 7.5.

The use of synthetic androgens, SARMs and the massive utilization of some of the T ester-based drugs derived from the initial use of senolytics and muscle-building to an extensive indiscriminate use, more often than not unjustified for addictive, body-building objectives, including an inaccurate understanding of what maleness is, means and is manifested; this is precisely based on their more socially undesirable aggressive facets. Androgenic treatments may cause functional problems, but there is not a clear similitude or continuity between the effects of even long-term TRT without medically advised anabolic use [462,463]. These drugs have been intensely studied, developed and often used without safe knowledge. Unfortunately, this situation continues and keeps expanding.

There are few doubts about the danger the use of anabolizers may carry for the cardiovascular system [464,465] and the brain [466], including behavioral and psychiatric disorders [467], but also extreme musculation, metabolically distorting bodybuilding [313], and obviously, alterations of the sexual (and reproductive) functions [468,469]. The extended lack of differentiation between androgen classes may be, in part, a justification for the confuse hoarding of all androgens in a single group of anabolic hormones (or anabolic drugs). However, anabolizers could not be easily equaled to the natural hormones, at least in their function-effects at the doses and paths of administration used, but especially with respect to hormonal regulation and a number of essential biochemical pathways that are alien to the most used drugs. Nevertheless, not all is negative, since some androgen-derived or related anabolic drugs (including SARMS), represent a fully productive pharmacological line, to treat common senile- or malnutrition-derived disorders, such as sarcopenia and cachexia [470], which continues to be developed at present.

### 7.4. Justification, Expectations and Variability of Long-Term “Testosterone” Treatments

The main focus of the initial TRT was largely to restore hormonal functionality of adolescents (and adults) with endocrine or reproductive disorders [471], but it was extended (often out of the clinical milieu) to “return” the libido and youth memories of sexual functionality to old men [472]. The prevention or correction of sarcopenia is an important application that is also being actively developed [301]. The use of T in the maintenance of physical health and psychological well-being came in later (despite being just the core of the pioneer experiments), and remains the most common reason exposed for their extended (and often uncontrolled) use [473,474,475]. However, the question over the effects of TRT on depression [476,477] is not closed, despite being an issue that is probably deeply influenced by the methodology and characteristics of the treatment and its high social impact, aggravated by our not yet sufficient understanding of the biochemical basis of depression.

Most of the substitutive hormonal treatments using androgens have been carried out without a sufficient knowledge of what androgens are and what they can do. This is especially applicable to the properties of the molecular species used, the duration of treatment and effects on natural T and other hormone functions, and their conjoint regulation. An example of this attitude is the extended confusion between T and AcT formulations, in a large part of the clinical literature, which assume unabashedly that all T-containing (free or esterified) pharmaceutical preparations used for TRT are simply galenic-equivalent vectors of T. This opinion is widely shared, and many of the meta-analyses, reviews and position papers on TRT compare results and conclusions but do not discriminate (or, even describe [478,479]) which specific molecular species, and/or via for their administration, induce the differential results observed [463,474,480,481,482]. In a few studies, however, the dose was modulated according to partial results, thus, helping prevent further complications [483].

Nevertheless, most studies and reviews on TRT show similar effects (largely those presented in Table 2), and are also favorable to the use of TRT in men, hypoandrogenic because of their age or other metabolic disorders. The idea behind the use of AcT instead of T (described in Section 7.3) is to facilitate and increase the regularity of treatments. However, this use of AcT does not take into account that T is not the “only” important androgen in the (male) body. The sizeable production of KT’s, the higher AR-binding capacity of DHT and the short half-life of T in plasma are factors essentially ignored in most of the clinically-oriented literature. Similarly, the androgenic environment modulation of patients under TRT is essentially unknown because of the generalized absence of markers, molecular comparisons of gene activation, binding, transport and disposal of the androgens given. Any comparison with the effects on endogenous androgens T, KT and DHT (but also including E2) is again absent. Unfortunately, most of the comparative studies on androgenization as a medical treatment seldom consider these questions, and neither check the comparative potency, toxicity, disposal and spectrum of metabolic effects of the “testosterone” drugs used.

The assumption of pharmacological equivalence is essentially based on the control observations of increased T levels after months or years of substitutive treatment using T or exogenous AcT. The T newly found in the blood of TRT patients is directly assumed to be formed via ester hydrolysis; this is responsible for the androgenic effects observed in the patients. However, this critical point has not been proven so far, and we do not even know how the synthetic esters of T affect the dynamics of DHT, KT or E2 synthesis in addition to that of T itself. Evidently, using long-lasting injections (i.e., providing a maintained flow of T for months) necessarily alters the natural mechanisms of control of T via the HHG axis, a factor that in most cases is simply ignored, practically relying only on checking the circulating T (only T) levels. This is probably the main disruption caused by the non-discriminate use of standard (albeit undefined) TRTs. The continued exposition to T release may produce serious alterations in the regulation of FSH and LH secretion, which may result in a chaotic regulation of the closer T-derived hormones (E2, DHT), which roles should necessarily participate in the normalizing physiological response to any TRT.

The rise in T elicited by TRT seldom takes into account the circulating levels of the same AcT injected, which are supposed to quantitatively yield T via esterase hydrolysis. The analysis of AcT, including the T esters used as drugs, is technically feasible [152,484], and injected AcT have been detected unaltered even in dry blood stains [485]. However, the information on their dynamics after or during treatment in blood, tissues and excreta is generally unpublished (or unavailable to the scientific community). The practical absence of molecule-to-molecule effectiveness comparisons, together with the absence of quantitative analyses, and the additional “corner-cutting” or unjustified generalizations when comparing different times, schedules, patterns, doses and the use of pre-established mechanisms for comparison, severely compounds the problem of endocrine/clinical evaluation of the effectiveness of deep intramuscular AcT injections for TRT. 

The time-honored precaution of using a washout period after a short-time treatment with drugs or hormones, such as T or AcT, is seldom (and inadequately) followed [486], and for logistical reasons, cannot be applied to large oleome-depot doses of AcT. The practical question of commodity of treatment: such as a few injections for a long time, has taken hold, despite the need to use safety systems [482,487]. The application of known pharmacokinetics of T and AcT used to apply different modes of administration [488] often are not included in detail (and neither checked) when planning or applying a TRT. This also goes for the safe use of washout or recovery periods interspersed along time in long-term drug treatments [489], but also used for long-term hormone treatments [490,491], and even for short-term TRTs [492]. The use of trans-dermic T for alternate months (i.e., no treatment in the *washout* period) was found to produce the same expected physiological responses with a halved T dose over the total time [493].

### 7.5. Effectiveness and Insufficient Overall Analysis, of Long-Term Use of AcT for TRT

Table 2 compares the effects of long-term administration of T and the main AcT drugs in use. It is difficult to try to assess whether there are real differences between treatments using T or its “preparations”, largely those using esterified T, since the populations of subjects (mostly adolescents or adult men) studied. These variables include age, disorders, presence of controls, use of a placebo group, purpose of the study and methodology used, size of the sample, molecular species used, dose(s), way of administration, duration, pattern and evaluation/safety parameters analyzed. These variables are compounded with the assumption that T is the only molecular species responsible for the effects induced by any of the hormonal precursors or agonists using an unproven premise that has been maintained for more than four decades. Despite its obviousness, this aspect has not been challenged, instead centering most of the interest in obtaining more effective and safe preparations [494,495,496].

**Table 2 ijms-23-11952-t002:** Main effects on men of prolonged “testosterone” replacement treatments using either T or AcT.

**Main Effects** **Observed**		**↑** **Testosterone**	**↑** **Glycaemia**	**↓** **Insulin** **Resistance**	**↑** **Insulin** **Sensitivity**	**↓** **Body Weight**	**↓** **Body Fat**	**↑** **Ipid** **Oxidation**	**↓** **Circulating** **Lipids**	**↑** **Body** **Protein**	**↑=** **Muscle-Mass**	**↑=** **Mobility**	**↑** **Cardiovascular** **Function**	**↓** **Liver** **Steatosis**
**H**	**V**	**T**	**GLUCOSE**			**LIPIDS**				**PROTEIN**			**SYSTEMS**	
T	*Tr*	[335,339,474,497,498,499,500]	[325,335,339,501]	[325,335,339,501]	[325,335,339]	[325,497]	[324,325,326,497,498,499,502,503]	[110]	[325,335,501,504,505]	[339,497,498]	[479,499,500]	[497,502]	[506,507,508,509,510]	
	*Inj*	[388,511,512]			[318]		[318]		[318,512]				[512]	
	*Or*	[513]												
TU	*Tr*	[514]												
	*Inj*	[327,494,514,515,516,517,518,519,520,521,522,523]	[328,340,475,514,515,516,524]	[340,515,522]	[329,330,340]	[328,516,517,519,521,522,524,525]	[327,328,329,330,475,516,517,523]		[31,327,328,475,514,515,516,519,524]				[524,526]	[517,523]
	*Or*	[523]					[527,528]							
TE	*Inj*	[529,530,531,532]	[533]			[533]	[520,533]		[529,531]	[534,535]	[307,532,536]	[537]	[509,538]	
TC	*Inj*												[509,538,539]	
UN	*ND*		[540]	[540,541]		[540,541]	[540,541]		[540,542]			[482]	[463,478,479,541]	
	*Tr*	[317,543]	[514]		[317]								[544]	
	*Inj*	[525,533,545,546]	[339,525,547]	[339,525]			[492,522,525]		[525,526,548]	[328,525]	[549]		[329,515,539,550,551,552,553]	
	*Or*	[554]	[528]											
**Main Effects** **Observed**		**↑=** **Bone** **Function**	**↑** **Kidney Function**	**↓** **Erectile** **Ysfunction**	**↑** **Sexual** **Function**	**↑** **Libido**	**↓** **Fertility** **Contraception**	**↑** **Life** **Expectancy**	**↓** **Depressive States**	**↑** **General** **Well-Being**	**↑** **Sleep Time**	**↑** **Hemoglobin**	**↑** **Hematocrit** **Hematopoiesis**	**↑** **Serious Negative** **Effects**
**H**	**V**	**SYSTEMS**		**REPRODUCTIVE**				**LIFE AND BEHAVIOR**				**BLOOD**		**NEGATIVE**
T	*Tr*			[326,497,513,555]	[325,326,473,474]	[325]		[507]	[474]	[473,474,512]		[497]	[500,556]	[311,462]
	*Inj*						[388]						[511]	
	*Or*												[[513]	[462]
TU	*Tr*							[557]						
	*Inj*	[523,558]	[518]	[475,559]	[476,521]	[559]		[507,523]	[476]	[475]		[327,494,519,526]	[494,518,556]	[556]
	*Or*			[430,528]	[560]	[430]							[500]	
TE	*Inj*	[561]		[533,535]	[535]		[438,520]	[538]	[507]	[329]			[529,556]	[556]
TC	*Inj*	[562]						[538,563]	[507]	[549]		[564]	[556]	[556,565]
UN	*ND*			[482,541]	[482]		[482,566]	[478,541]	[482,567]			[542]	[482,541,542]	[480,508]
	*Tr*												[317]	[568]
	*Inj*	[543]		[545]	[545,551,569,570]		[388]	[546,551,563]		[549,571]	[492]	[564]	[565,572,573,574]	[550,565,568,572,575,576,577]
	*Or*													[568]

The numbers shown in the table are references to studies of the effects of T treatment in men. These studies show, as principal or significant, most of the effects listed in the first row (in addition to further information and other effects not so generally acknowledged). Thus, the top row (vertical descriptors) shows the main effects observed in most of the studies (↑ meaning an increase or improvement, ↑= maintenance within normalcy, and ↓ decrease or reduction in the parameter, which may represent an improvement or not). The effects generally considered positive are shown using a pale green cell background, and pink has been used for those considered negative or potentially dangerous. The effects have been loosely grouped according to the type or subject of the function described. The last column/section, NEGATIVE, corresponds to other varied, general or undetailed negative effects of the treatment (mainly cancer or cardiovascular disorders). Hormone (column H): T = testosterone; TU = T undecanoate; TE = T enanthate; TC = T cypionate; UN = undefined/unknown T derivatives, formulations or combinations of androgens used in TRTs. Mode of administration/via (column V): ND = not described, or combinations of ways (i.e., in the context of meta-analyses or reviews); Tr = transdermal: gels, patches or other similar procedures; Inj = injection, largely deep i.m.; Or = oral administration.

The wide dispersion of data and results available for TRT is not only a consequence of the polyphyletic origin/purpose of the large number of studies conducted so far, but especially of the unproven assumptions ruling the “practical clinical” use of androgens. There are a considerable number of comparative studies and meta-analyses on the effects of TRTs, but comparisons are difficult because of the reasons explained above that can be summarized in three main problems affecting homogeneity: (a) what is actually known as “testosterone” by the authors? (b) the variability in extension, objectives and methodology (duration, dose, mode of administration) and (c) the absence of periodic patterns of administration as a way to protect the integrity of function of the HHG axis. 

Thus, I decided to just list the articles directly available, and pair the papers with the effects most commonly described of androgens in the series studied. This is in line with other previous analyses [474,482]. The most common types of androgens used were differentiated in the analysis: T and the AcT. Most of the T studies consulted were carried out using gels and patches (largely short-term), whereas most of the AcT treatments were long-term, and largely referred to injections of—mostly—T undecanoate. The studies using T were essentially transdermal, short-term and oral, inducing mostly positive responses, whilst the effects of AcT—largely injections and long-term—were also effective, but, with respect to those of T, there were different lists of results, now including some negative effects. These included circulating lipids, cardiovascular disorders and contraception, but also included positive influences on the widely discussed issues of T incidence on life expectancy and depression.

Since the data were not statistically comparable, a crude analysis was carried out, relying on the interpretations of the conclusions presented in the papers cited (many of them already being meta-analyses and reviews). The number of citations sustaining the different effects described were counted irrespective of the mode of administration and a rough ratio of T vs. AcT was estimated for each entry after correcting for the total number of references counted. The following short list presents the parameters analyzed (only those with a minimum of data, or those including several other effects), listed according to the number obtained from this ratio compared with T. In green, the “positive” results are shown, and in red the “negative”; the up arrows represent an increase (or improvement), and the down arrows show a decrease (or worsening condition).



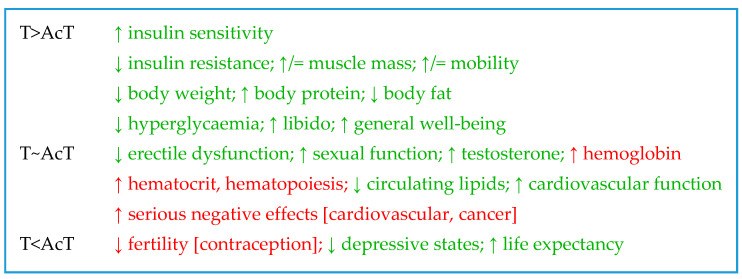



A large part of the AcT differential pre-eminence on some effects may be a consequence of the extremely long periods of treatment, the use of high overall doses and the obvious alterations of the physiological regulation of androgen levels and functions. These problems also affect the transdermal procedures used for T, but the TRTs duration is often counted in weeks or months (far away from the months or years of most AcT-based TRTs). In any case, there is a graded difference in the type of effects observed between T and AcT, despite their coincidence on questions related with sexual functions and the effect on circulating T. This superficial analysis shows, however, that the references in the studies analyzed to issues more related to energy metabolism, such as insulin control (and glycaemia) and body protein, were more frequent when describing the effects of non-esterified T. Thus, the results obtained for AcT were—probably—a consequence of the released T rather than direct effects of the larger ester molecules. The apparently marked beneficial effects of transdermal T on insulin resistance and the increase in insulin sensitivity, which were necessarily observed under relative short-term treatment (gel), can be directly related to the implication of E2 in the control of glucose metabolism (and lipid oxidation). This difference hints to the facility of aromatization of abundant free T to E2 as a critical control node of glucose (3C) and 2C energy metabolism, as previously reported [3].

Injected AcT depots release T, but probably not in the ideal (i.e., *circa*-T endocrine homeostasis) conditions to sustain a sufficient and controlled flow of E2 to maintain energy partition and increase lipid oxidation, as deduced from the different profiles on E2-dependent effects of free T and AcT administration. However, this flow did not hinder the conversion of T to DHT, which may take place even when T remains esterified [578]. The main differences in clinical effects between both types of treatment, however, were those derived from cardiovascular protection/damage. Almost not cited for transdermal T, they were represented both in favor of protection and as possible causes of serious ill effects in the AcT group [511,542,556,573,579]. The differences were, again, more marked between T and AcT for the TRT effects on contraception—an effect sought for long-term treatment [388,438,520] with often uncertain results [580]. Notwithstanding, infertility may also be an unwanted consequence of TRT [482,529]. The application of T as a male contraceptive has been used for a long time [447,581,582], but nevertheless, remains a hormonal option despite some shortcomings and relative unreliability [581,583,584].

There are, also, differences on hematopoiesis between the two TRT groups described. Long TRTs seem to clearly interfere with erythropoiesis, essentially by increasing erythropoietin activation and inhibiting hepcidin [585,586], which results in higher blood cell counts, hematocrit and hemoglobin. These effects are patent in long-term TRT using AcT in comparison with T gels [587]. Nowadays, warnings about these negative blood alterations are already present in direct access information on TRT products [578], and are taken into account when planning androgen substitutive treatments. The T effects on polycythemia do not require aromatization [588], which can be assumed to be not mediated by E2.

The analysis presented above is not quantitative and has no statistically endorsed reliability, but is the narrative sum of a fair number of widely different sources. A main reason for using this low-certainty procedure is the generalized absence of unifying criteria, which, however, have the advantage of preventing the (common and selective) exclusion of many studies, thus, adopting a wider encompassing (albeit less defined) vision. The studies analyzed were carried out using quite different criteria, methodology, focus, subjects, duration and even types of molecules or combinations. In addition, they were developed under different contexts of knowledge (along several decades) on the function, use and analysis of steroid hormones and the pathologies they were used to limit/prevent.

Nevertheless, the number of studies included their size, criteria of selection, and, especially, the availability of the reviews and analyses are important aspects to be taken into account, in part as “negative” factors that may affect the relevance of the present analysis. The obvious caveat of precisely using the specific papers cited is the logical assumption that any other selection may result in different (albeit also provisional) conclusions. However, the number of studies analyzed their varied “age”, context, origin, purpose and widely varied focus (plus the exclusion of animal-based studies as a rule), which may help to set the context for the practical usefulness of the limited results obtained: hints, at the most. The fact that there is a notable coincidence in the summary of effects with previous studies [482] also helps to show that only the most apparent common ground has been highlighted. The observations are necessarily limited, but nevertheless clearly indicative of the existence of differences in the effects elicited by using T or its esters.

Continuous treatment with AcT alters the hematopoietic system in a way apparently not (or, possibly, not yet) detected during short-term (and largely transdermal) TRTs using T [587]. Since a fair number of androgen derivatives elicit porphyrin (and hemoglobin) synthesis and erythropoiesis [589,590], the differences between T and AcT may hint at natural AcT implication on this aspect of metabolism, but there are no specific data to sustain this appreciation.

The effects observed on body composition and circulating lipids may be a consequence of the normalization of circulating T levels, which also help explain the improvements in sexual arousal and function. Nevertheless, the present analysis does not support a clearly different functional role for the physiologic AcT and T, other from those related to E2 production and AR binding. It has been postulated that AcT could act as a temporal storage (limited in mass) for T in tissues [60,153,591]. Thus, the function of AcT at physiological levels may be not quite as different from that of T, as are the widely different functions of estrogens and their fatty esters [9]. This is in agreement with the variability of physiological fatty acid moieties in the small pool of natural circulating AcT [437].

It is fundamental to obtain a critical and systematic analysis of the effects of the different molecular species used for TRT. At present, we already have available (and rapidly developing) the powerful methodology to effectively analyze the kinetics of utilization of these drugs and to establish their interaction with other androgens (and other classes of steroid hormones). The overlapping, limited, selective TRT studies available often repeat incomplete analyses, or select very specific targets in highly constricted conditions, leaving large (and repeated) gaps in our knowledge. Consequently, they do not help enough to consolidate what is known. This discontinuity in the research focus is fully in line with a recent general comment on the clinical studies’ repetitiveness and focus inconsistence with respect to previous analyses [592].

The higher lipophilia of AcT used in pharmacological preparations is a bonus for the smooth absorption and extended effects of T [408] whilst preventing its immediate inactivation via glucuronic acid conjugation. Notwithstanding, the sustained use of AcT drugs for long-term anabolic, restitutive or simply senolytic actions, must come with two important caveats to control and/or take into account: (a) the need to watch the red blood cell and hemoglobin levels [573,593]; and (b) the deep and long-lasting interference of a continuous and persistent supply of T, from deep intramuscular injections on the control of the HHG axis, deregulating the levels, function and adjustment of both T and E2 (and indirectly, those of DHT, other androgens and gonadotropins) [594,595].

### 7.6. Additional Questions Posed by TRT in Women

So far, a large majority of TRT studies have been centered on men, but there are a growing number of studies dealing with the therapeutic use of T in women [596,597,598]. The ease with which T is aromatized to E2 is a fundamental aspect to consider when using TRT in women, despite their quantitative needs of T being apparently lower than in men. Nevertheless, this question has to be taken into account because of the danger posed by the possible potentiation by E2 of estrogen-sensitive cancers [599].

A key question, however, is the indiscriminate use of the polysemic term “androgenization” applied to women, both to POCS patients [600] and postmenopausal women [601]. There are marked differences in the discrimination, definition and, consequently, pathognomic explanation of the POCS dual disorder: ovarian or adrenocortical [602]; however, androgenic dysfunctions [603] and insulin resistance [604,605] are common occurrences in both, including the superimposed appearance of menopause on POCS patients [606] The similarities of POCS with MS have been also associated, too, with common pathogenic manifestations 

The question of androgens in women is even more complex than in men, especially because of the relatively new finding of KTs, hence, increasing the adrenal cortex importance in supplying a steady flow of androgens to women [13]. Androgens in women are essential, for functions not necessarily coincident with those of T’s full spectrum of actions). The blood levels of androgens (as a whole) are fairly higher in women than those of estrogens (even when he KTs are not taken into account [607]:DHEAS > DHEA > A4 > T > DHT

At least 25% of T is produced in the ovary, 25% more in the adrenals and the rest is the consequence of peripheral interconversions (cited by [608]). The ovary does not cease its production of T in women for at least one decade after menopause [79], but the overall production of A4 by the adrenals decreases markedly, which converts T in the main androgen produced overall (as compared to A4) [609]. The global body production of T in post-menopause compared with the reproductive period decreases by one third, whereas those of A4 and DHEA are halved [608]. In POCS, there is a higher thecal secretion of A4 [609], increasing the androgen levels in adrenal glands [610], but the conversion of A4 to E1 is enhanced, thus, further decreasing A4 availability [611].

In ovarian vs. peripheral blood, oophorectomy decreases the levels of T and E1, but not those of A4, E2 and DHEA [79]. The faster turnover of T, however, could not be sufficiently compensated by a fairly maintained T production at menopause and further on. A comparison of androgen and estrogen levels in ovary blood vs. systemic blood [612] showed a 15-fold decrease in T levels compared with “only” 4-fold for A4 and 2-fold for E1 and E2. This may explain the quantitative importance of A4, but also, probably—and mistakenly—includes the dubious “androgenic” contribution of the fairly maintained levels of DHEA to the “androgenization” often described for old women. It has been found, however, that 11-keto-androgens are the main adrenal androgens in PCOS [613]. Thus, despite the marked adrenal decrease in women in A4 production with advanced age [614], the decrease in T at menopause may have a more effective result when lowering the androgen availability of postmenopausal women. There are other possible explanations to the post-menopausal androgenization, such as aging [614] and the manifold increase in the possibility of developing MS [615] when the HHG-elicited production of E2 falls at menopause, helping spur obesity and finally equalizing older women with mature men’s long navigation through MS. Increases in T can be also attributed to cancer and other metabolic disorders in the postmenopausal period [616,617]. We lack information about the roles played by T, its aromatase conversion to E2 and the unexplained possible function of excess availability of E1 in this critical period. In fact, its association with obesity [618], and the adipogenic effect observed in rats [619], may help explain in part the shift in lipid metabolism at menopause, and its leaning towards a MS situation.

The changes in the hormonal makeup of women after menopause may also give more transcendence to the modulation of KTs and their closely associated corticosteroids in the adrenal cortex (Section 5.3). The fall in tissue E2 and T availability is probably directly related to a shift in the adrenals, with a lower production of classical androgens and more marked corticosteroid effects [620] including higher circulating cortisol [621]. However, the treatment with DHEA helps limit the corticosteroid effects [622]. These data agree with a shift of the adrenal cortex towards the synthesis of corticosteroids in the detriment of androgens.

Menopause also induces a change (to worse) in the maintenance of bone health [623,624], largely due to the fall in E2 and the decrease in its support of bone homeostasis. However, this could also be due to increased glucocorticoid activity [625], since glucocorticoids promote the development of osteoporosis in animal models [626]. Osteoporosis is common in post-menopause [627,628], and is often found to be associated with obesity [629,630], even with sarcopenic obesity [631]. The implication of a wide number of regulatory mechanisms, such as the relationship of post-menopausal bone health with bone marrow adipocytes, have been studied [632]. The use of E2 replacement protocols helps correct the danger of osteopenia/osteoporosis [633,634]. The metabolic and hormonal turmoil of menopause also affects brain function [635], protein pool maintenance [636] and the combined consequences of a lower availability of T and E2 on senescence, sexual functions, behavior and well-being. These effects can be partly counteracted with T and E2 replacement therapies. There are, yet, few studies on substitutive androgenic treatment in women for the improvement of the already patent T deficits in post-menopause [596,598,637] or heart conditions [638]; the dual therapy of T and E2 have been already tested [639]. However, the weight of misconceptions and lack of acknowledgement of the crucial metabolic regulation role of T in all humans, irrespective of gender, somehow restrain the full clinical use of T for the treatment of disorders related to menopause (and often compounded by MS) in women [598].

The duality of T and KT (and 4A, probably to a minor extent) as key female pre-menopausal androgens, and their coincidence (or not) in functions, needs to be more specifically studied, as it is a critical point for the eventual effective androgenic treatment of women with hypoandrogenism. This includes the clarification of the (differentiated) path for DHEA treatment, which beneficial effects are well known. However, we still need to understand why so much T is produced, and also, why it simply seems to vanish (unused?). Any sustainable explanation needs to seriously take into account the third party in this steroid hormone age-related disorder: corticosteroids. Unfortunately, we do not have enough concrete information on the interconnection between corticosteroids and the basic genetically coded processes of senescence and age-driven systemic-obsolescence (with T and E2E2, as explained above). These processes have evolved to determine the inherited functional protocols on which stand the survival of our species, and are hardly subjected to rapid modifications along a few generations.

In summary, hypoandrogenism in men is usually related to aging and/or the development of MS [576,640] and related disorders. The key factors being an insufficient production/availability of T, and consequently, an even deeper insufficiency of E2. In women, however, menopause may induce a different kind of hyperandrogenism, characterized by a limited E2 availability largely due to insufficient T, not compensated by the lower A4, but with higher proportions (but not levels) of DHEA; all of this context is paralleled by higher E1 over E2. The assumption of androgenicity in old women may be caused mainly by the interpretation of relative data: total androgen (largely indexes, not levels and no discrimination of molecular species) rises (over pre-menopausal conditions) with respect to E2, since E1 predominates, and the drop in circulating T (and to a minor extent A4) is “compensated” by lower changes in DHEA (misinterpreted as a full androgen, in the testosterone sense). Thus, actual female menopausal hypoandrogenism is largely hidden behind a screen of indexed data, and occasionally true hyperandrogenism caused by other pathologies. Senescence in women, thus, reverts to the same problem that hypoandrogenic old men endure: insufficiency of both T and E2 (playing havoc on glucose and lipid metabolism: Section 5.1 and Section 5.4); a situation mimicked in the Turner syndrome [641], induced essentially because of low T availability.

## 8. The Use of Androgens for the Treatment of Functional Disorders

The substitutive administration of androgens to hypoandrogenic men is conceptually similar to that of estrogens to women, in which the ovaries are no longer functional (for ovulation, but not always for endocrine secretion). However, the sex differences are critical: the deficit (of T and E2) affects not only old men, but also much larger (commonly younger) groups of men [385,387,642] and women [386] with MS. The planned obsolescence of the HHG axis starts sooner in men than in women, but both genders tend to show similar situations in advanced age, despite a slower long progression in men vs. an abrupt acceleration in post-menopausal women. The feeling of unexpected loss of functions already experienced by mature (i.e., not yet old) men produce deep changes in their attitudes, intending to maintain (or recover) their intrinsic maleness. This ancestral fear has been always tried to correct, to no avail, mainly because of a lack of knowledge and adequate means. It affects other animals, especially those societies with alpha-males hoarding a number of females (harems) [643]; these situations invariably result in the challenge and substitution/demise of the old patriarch [644]. Testosterone levels and dynamics play a critical role in the maintenance of this reproductive selection system [645].

Despite the caveats and preparation of Medical Societies-endorsed position papers, the normalized use of efficient TRTs is far from being established. There is a constant evolution, reflected in the short-time between actualizations of these statements [481,646,647,648,649]; but the use of TRT for aging (and/or MS-derived hypogonadism) keeps growing [650]. The focus is now shifting to the discussed and controversial use of TRT in women [651,652,653], which may be related to the deeply ingrained (and scientifically unjustifiable) notion that sex hormones fully imply an “absolute” link to gender: androgen for males and estrogen for females, with no space in between. This unproved belief has been a pillar of “sex hormone medical knowledge”, and remains true up to the present for generations of physicians, scientists and social leaders and activists.

To my knowledge, the KTs have not yet been used for androgenization treatments in the general clinical practice to help maintain steady and effective growth/tissue maintenance and the basic androgen functions of metabolism. However, they may be called to—at least in part—substitute T (and synthetic AcT) in these tasks with a higher degree of safety. Their antipodal relationship with GC has yet a dearth of potential applications to be understood, tested and developed.

The contemporary massive use of androgens as senolytic agents has been re-initiated by gerontologists, but now it has extended to other areas of medicine, being finally (but reluctantly) accepted by the endocrinologists probably because of the consequences of their generalized (and often uncontrolled) use [654,655], and especially, because part of the ravages of old age and MS are clinically improved by TRTs [656,657]. In addition, TRTs also help regularize glycaemia [658], maintain muscle mass [659,660], improve well-being [661] and reduce the development/severity of cardiovascular disorders [662]. However, the uncontrolled use of androgenic drugs has been observed to induce serious health problems, as described in Section 7.3, including psychological and personality alterations [663] and even reproduction-related loss of function [664,665], including infertility [390].

A large number of old people show a marked loss of body protein in special muscle mass [666,667,668], often compounded with osteopenia [669]. These changes fit in the widely extended sarcopenia spectrum [670,671], inducing or compounding frailty [672]. These changes are commonly associated with disorders grouped under the metabolic syndrome umbrella, and/or aggravate previous disorders and pathologic chronic conditions [673,674], globally enhancing the severity of frailty and increasing the dysfunctionality, morbidity, dependence and mortality of the subjects [675,676]. The changes in body composition (affecting mainly protein), decreasing turnover of structural proteins [677], and thus, reducing their reposition and functionality, do not necessarily affect body lipid stores. Thus, sarcopenia can often develop while maintaining obesity (e.g., sarcopenic obesity) [678,679]. The obese can become sarcopenic and frail while maintaining their main energy (TAG) stores full, with a largely unused mass of fat which constitutes an additional burden [680]. A key metabolic difference between sarcopenia and cachexia is the indiscriminate use, in cachexia, of all body substrates susceptible to be used for energy (fat or protein), [681,682]. TRTs have been also used to complement the diet, using an adequate (supplementary) supply of energy, protein and micronutrients, as a way to limit the ravages of malnutrition [683,684,685], sarcopenia [300,301,686] or even advancing cachexia [687,688].

In men, old age by itself tends to correlate with lower levels of T [332,689], but often maintains normal or high levels of gonadotropins [690,691], helping suppress the gonadal production of T [691,692]. This situation suggests that hypoandrogenism in older males is a probable consequence of the HHG axis dysregulation. The deficit in estrogens in women can be treated via the substitutive administration of estrogens (often combined with progestogen) [693]. This may help to correct the problem of the E2 deficit, but not that of T, which could not be improved this way, since the aromatization path is not reversible. In people enduring gender reaffirmation processes, the exogenous hormonal treatments tend to overwhelm the (often diminished) physiological synthesis of either T (for men [694]) or E2 (for women [695,696]); therefore, at least during the transition phase, the HHG regulative system is expected to be largely inoperative because of signal saturation blockage. Hormonal treatments may be effective without the need for additional modulation using gonadotropins or GnRH [697]. 

There is considerable (recent) literature on gender-normalization endocrine handling, but in male reaffirmation procedures, the basic treatment is similar to the long-term hypoandrogenism situation described in Section 7.5, i.e.; that the administration of T or AcT helps the masculinization of brain [698], restores musculation and functional re-adjustments [699], as well as inducing the manifestation of secondary sex characteristics [700,701]. In affirmative male-gender men, T treatment does not exacerbate a pre-existing POCS disorder [702], but increases endothelial inflammation [703,704]. In reaffirming women, the specific suppression of T synthesis may be even eliminated from the treatment because a high part of the naturally produced T is largely aromatized to E2. However, the use of T enanthate for the hormonal reaffirmation of men [449,705] may help explain the absence of the expected rise in E2 [706]. The use of T in gels [707] or subcutaneous injections has been found to work better in practice for this purpose [425], probably because T can yield E2 more easily than AcT.

When discussing the benefits of TRT to patients with hypoandrogenism, it is critical to think of T not only as an androgenic vector, able to regulate metabolic functions, maintain muscle, cognitive functions and the diverse functions sought to recover, but that T is also the main precursor for E2 synthesis. Consequently, E2 has a pivotal role in energy partition, anti-oxidative functions and for the cooperation with T itself, and even GC, to regulate substrate utilization. The beneficial effects of T on MS rely not only on T itself but on the cooperative muscle–bone–liver–brain interactions between T and E2, including the control of fat deposition and its disposal. This dance-duet equilibrium is fundamental to recover full metabolic homeostasis. Steroid hormone treatment of a metabolic disorder (or a constellation of them, such as MS) should consider this special relationship.

In short: the estrogen collaboration to TRT may help considerably in the adjustment of metabolic energy partition [3,9,18], which directly affects body fat stores [708,709,710], protects the heart [711,712], maintains muscle function [713,714], exerts neuro-protective effects [715,716,717,718,719] and regulates mitochondrial integrity and function [35,720,721], in part because of E2 anti-oxidative effects [35,722,723]. E2 is also necessary for spermatogenesis in the testes [724,725]. The main metabolic function of E2 is probably its role in the control of energy balance and substrate energy partition, in part by controlling insulin action in the regulation of glucose metabolism [3,9,726].

There is enough basic and clinical knowledge to develop MS treatments based on the careful correction of its main hormonal deficit, T. However, care must be taken to control the oscillations of T (and E2) levels along the whole treatment, interspersing—necessarily—periods of rest or washout (i.e., no hormonal treatment) sufficient to maintain (and prevent blocking) the periodicity of gonadotropin secretion and protection of endogenous testicular endocrine (and reproductive) function. The widely used avenue of the substitution of T by AcT has shown—so far—only limited effects on glucose levels (Section 7.5), compared with the consistent inverse relationship between T levels and insulin resistance [219,727,728], with a direct incidence on MS and body lipid handling [727,729,730]. The differences between T and AcT hint at the T effect being probably mediated (at least in part) by E2, especially if we compare the known effects on lipid handling and body mass elicited by estrogens and acyl-estrogen SERMs [3,9,155].

Testosterone may be an excellent senolytic when, and if, administered reasonably: i.e., only when needed, and in a carefully balanced way, to just cover the deficit (with as few disturbances as possible) of patients of hypoandrogenism correctly diagnosed. This implies that repeated checks and controls should be maintained (at least) throughout the whole treatment to prevent unwanted effects. Nevertheless, utmost care must be taken on the clinical use of androgens, with the focus set on the individual conditions of the patients. The preservation of muscle mass and cognitive functions should be the main objectives of TRT due to their critical role in the maintenance of the quality of life.

Box 1 shows a list of the main factors to consider when planning or developing a pharmacological intervention to treat hypoandrogenism, such as is presented in senescence and MS. The points described are simply a summary of most of the questions treated along this review.

Box 1Aspects to consider in the design of a substitutive androgenization treatment.Any plan to carry out an androgen substitutive/compensating treatment should consider, at least, these aspects:
*Patient conditions.*First, the real existence of a need for androgenization treatment must be clearly established after an intensive and exhaustive metabolic and hormonal analysis of the subject. Then, the plausibility of carrying out an androgenization treatment must be decided upon, taking into account the individual conditions: sex, age, nutrition or hormonal disorders (especially including metabolic syndrome and its related pathologies, andropause, menopause, old age and hypogonadism). These aspects should be weighed according to the severity of the conditions, time available and health objectives sought. Psychological factors must be considered because of the probable duration of the process and the behavioral/social changes expected.*Molecular species used*Currently, practically all androgenization treatments are based in either T or a number of *eka*-T products, as described in the text, but their mode of action is not the same. This analysis should be open to specific (even combined) treatments including T, AcT, KTs and the fully synthetic drugs available at present (not discussed in this review), tailoring the molecules to the needs of the treatment (and the extent of the risks taken with their use). Evidently, age, sex and metabolic disorders are critical factors to ponder in addition to a solid reason for applying an androgenic supplementation treatment. The eventual addition of other hormones, such as E2, should be also considered.*Way of administration and dose*This aspect directly depends on the molecule(s) selected, the disorder to be treated, and the real rates of release/inactivation. The problems posed by the different methods of administration are known and should be better specifically known through more extensive and complete checks of the molecular species involved. In general, it is better to use repeated short term procedures (i.e., transdermal applications) to maximize the possibilities of adjustment, evaluation, control or interruption of the treatment as needed.*Duration of the treatment*Time of treatment is closely related to the two previous points, since most of the procedures used so far rely, essentially, on massive, uninterrupted and prolonged treatment with (usually) a single hormonal agent. However, hormones are extremely sensitive to physiological modulation and are carefully regulated in vivo. Only good feedback of the effects sought by the treatment may provide the sufficient level of safety and effectivity to carry on. This is another reason favoring shorter treatments (preferably repeated after safety interludes), and the continued control of physiological functions, hormone levels and related markers, as well as behavior/patient satisfaction (and compliance) feedback.*Maintenance of hormonal homeostasis*The administration of a drug with hormonal effects constitutes *per se* a destabilizing intrusion in the body humoral homeostasis, obviously, irrespective of the nature and quality of that homeostasis. In the case of “testosterone” long-term treatments, often the only guide available (differential effects) can begin to be observable after long saturation (resulting in inactivation) of the physiological control systems, after using high doses and a prolonged time. Only a few studies include periods of “rest”, interspersed after several months of treatment. This is a critical point that needs further (and deeper) development, extending the treatment periods based on limiting the overwhelming damage to the natural biological cycles (i.e., monthly, bi-weekly or even shorter alternate periods), and leaving time and space for the homeostatic systems (essentially the HHG and HHA axes) to re-start and help solve the disorder instead of enduring or fighting the powerful and continuous drug intrusion.


## 9. Conclusions and Perspectives

As in the case of estrogens [9], androgens are a group of hormones with quite different (often complementary) functions and modes of action, which are synthesized along a pathway (closely related to those of corticosteroids, and especially, estrogens) distributed in a few organs, strategically located to serve their multiple functions. They are all hormones acting not immediately, but essentially in the mid-term. This is accomplished largely (albeit not exclusively) via the modulation of gene expression, thus, their actions are of relatively prolonged effects, and are exquisitely regulated by a complex web of mechanisms to adjust their responses to the changing needs of the organism. They complement (and often also regulate) the shorter-acting hormones (e.g., insulin, catecholamines). Androgens (largely T) add a component of continuity and stability to hormonal regulation. This is the general trend, but in the specific case of testosterone, its half-life is much shorter than in most steroid hormones, a factor that allows for faster (bulk) responses (often non-genomic, and thus, also faster), but which extremely complicates its administration, and consequently, the maintenance of pharmacologically stable levels.

This is an open door for the treatment of MS and associated (or derived) disorders (obesity, type 2 diabetes, cognitive impairment, senility, sarcopenia, cachexia), but also includes frailty and the reproductive problems—about sex/fertility—often directly related to T deficiency. Most of the initial work in this field has already been carried out and is available (albeit disperse, and often forsaken) in the literature, thus, we need to widen our scope, learn/remember what is already known, organize and cross-check this huge amount of information with present day metabolic knowledge. When carrying out this actualization, we will be able to design and apply safe and adjusted procedures to help stop the increasingly extending epidemics of metabolic disorders which blueprint origin is within us, coded in our genes.This complex protracted decay situationcannot be solved using untested procedures of dubious effectivity, such as dietary manipulation or applying (untested) narrow-function drugs, instead of relying on new research and the sidetracked or forgotten scientific knowledge obtained by our predecessors in Science, an stored legacy which is (theoretically, at least) available to us.

An adequate focus of T substitutive treatments allows for the maintenance of T and E2 levels and functionality, but exerting a persistent blockage of the HHG axis and the brain attempts to keep the circulating T homeostatiss. Circadian variations in T have been described for men, in parallel to gonadotropins, E2 and other hormones [731,732,733], rhythms that tend to be less marked with advancing age and lower T levels [734]. However, ultradian cycles have been much less studied; there are sparse data showing a seasonal variation of plasma T in men, but the dispersion does not allow to clearly establish periodicity and duration [735]. Nevertheless, a study on 20 subjects showed the existence of several-day cycles (8–30 days, with most subjects in the 20–22-day range) in T levels in normal men [736]. Unfortunately, there is not enough information on this critical issue, since most TRT are prolonged from months to years without pause, necessarily interfering/destroying the physiological rhythmicity of the circulating T.

A number of genetically, epigenetically or man-made (e.g., diet, disease) disorders, such as MS or senescence-related hypoandrogenism, bring us again to the same questions (and provisional answers). All of them share a common distinctive alteration, such as the deficit of T (and, more or less directly related, deficit of E2); generated with the participation of the brain (hypothalamus) and our adjustment to life patterns established along evolution. These sets of life cycles seem hardly applicable to many present-day individuals (despite sharing the same physiological make-up as our ancestors of a few millennia ago). Public health, food availability, literacy knowledge, the structure of our society and many other advances, mark the difference in the survival of humans as species (however, not always in a positive way), drifting the massive reproduction/early death strategy by that of controlled reproduction/longer active life, obviously not without severe problems from the present environment (in a broad sense) and from our genes (i.e., planned obsolescence of men via MS, executed in part through a HHG-mastered dwindling of T (and E2) levels). These disorders affect hundreds of millions of individuals, and we do not yet have available and effective ways to cope with them. This may be a direct consequence of the genetic resilience that helps maintain our species to thrive.

There are sufficient pointers signaling the deficit of T as one of the most extended, repeated and mistreated medical problems. Thus, caution is needed, along with knowledge, to guide the use of exogenous T to correct metabolic-functional disorders caused by age, pre-established biological obsolescence mechanisms, a maladjustment to diet, stress, disease or a cumulative sum of the life-history of any human. This should always be taken as a palliative remedy, seldom as a cure for a settled or unstoppable disorder. It may help improve life conditions, extend the period of (worthy) life and compensate for the disorders observed because of the decreased availability of T. In fact, T is an extraordinary hormone, a true “elixir of life” [737], as was often defined by the pioneers in their study, only comparable to its life-peer E2; however, it is not a panacea, and does not work alone, but within a coordinated group of steroid hormones and controlling factors that induce the smooth changes expected from steroid hormones in both women and men along their entire life.

## Figures and Tables

**Figure 1 ijms-23-11952-f001:**
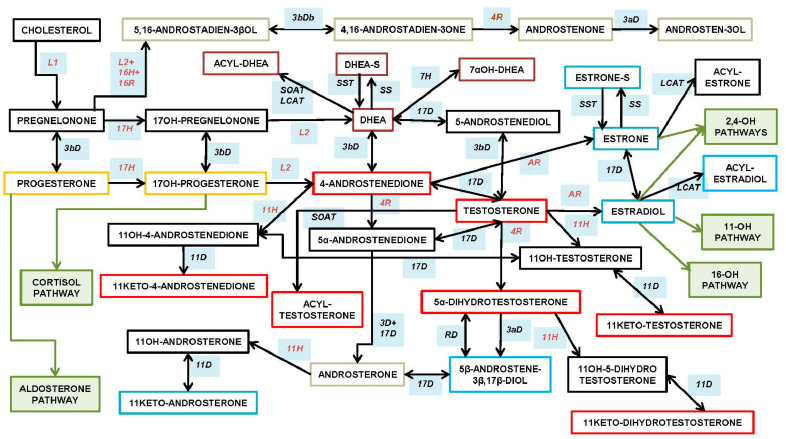
Main androgen sytntheis pathways. This figure represents the physiological molecular species secreted in/by adrenal glands (intermediate and cortical layers), testicles, ovaries and the brain, as well by a number of other organs or tissues with a critical participation in these processes (e.g., skin, liver, adipose tissue). Since the synthesis of androgens (especially in the adrenal glands) is closely related to the two parallel corticosteroid synthesis pathways (they share location and a few enzyme activities), the start of these paths has been indicated in green squares. This includes the estrogens, which metabolism is much more closely intertwined with that of the main androgens. The 16-[estriol] and 11-hydroxylative pathways, as well as the catechol-estrogen specific pathway, have been included only as annotations in green labels. Black arrows show the enzyme-driven changes between molecular species; two-headed arrows show reactions that are potentially bidirectional. The main androgen molecule borders are red, violet in those sharing androgen and estrogen capabilities and blue in the fully estrogenic molecules; the progestogen borders are marked in yellow, and the androgenic pheromone species are in grey. The remaining molecules (black borders) may show a limited (if any) androgen receptor binding ability. The enzymes intervening in the reactions depicted are listed below the figure. They are presented in borderless pale blue rectangles in contact with the corresponding black arrows; the letters are in brown for mitochondrial and black for microsomal (and other location) enzymes.

**Figure 2 ijms-23-11952-f002:**
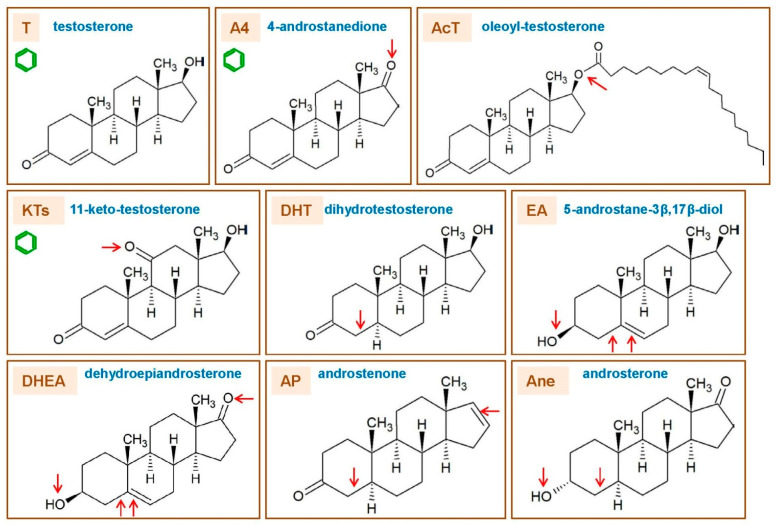
Types of human androgens. The androgen molecular species (or groups of them) show similar chemical structures, are synthesized in a number of different tissues (but mainly in adrenal glands and the gonads) and elicit physiological effects that are largely complementary. DHEA: dehydroepiandrosterone; T: testosterone; AcT: 17β-acyl-T, a group of esters; KTs: 11-keto-androgens (essentially derived from T, DHT and A4; the 11-keto forms are more active than the 11-hydroxyl ones); DHT: dihydrotestosterone; A4: 4-androstenedione; AP: androgenic pheromones; EA: estrogenic androgens (i.e., they bind to the ER and AR); Ane: androsterone—a catabolite of T—which is an agonist of the farnesoid receptor, acting in the regulation of bile acid signaling. There are many other androgen catabolism products and intermediate molecular species of the androgen metabolism, which specialized functions have not been described in depth, but have been studied as pharmacological subjects, metabolic markers or substrates for synthetic hormone production. The androgens susceptible of aromatization are marked with green benzene icons. Small red arrows point to the distinguishing structural features of the different groups of androgens in comparison with T taken as the standard and best-known androgen.

**Figure 3 ijms-23-11952-f003:**
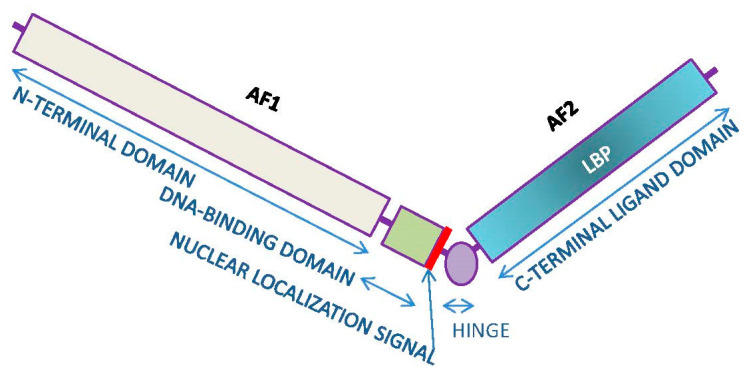
Functional structure of the androgen receptor (AR). The diagram shows the two distinct parts of the protein chain, joined at a flexible joint (hinge). The longest arm contains the N-terminal domain, incorporating the AF1 (activation function 1) binding sites, as well as the DNA-binding domain and a short sequence (nuclear localization signal) needed for the nuclear transport of the AR. The shorter arm (C-terminal domain) contains the AF2 (activation function 2) binding site. This domain contains a key binding niche, the LBP (ligand-binding pocket), in the core of AF2.

**Figure 4 ijms-23-11952-f004:**
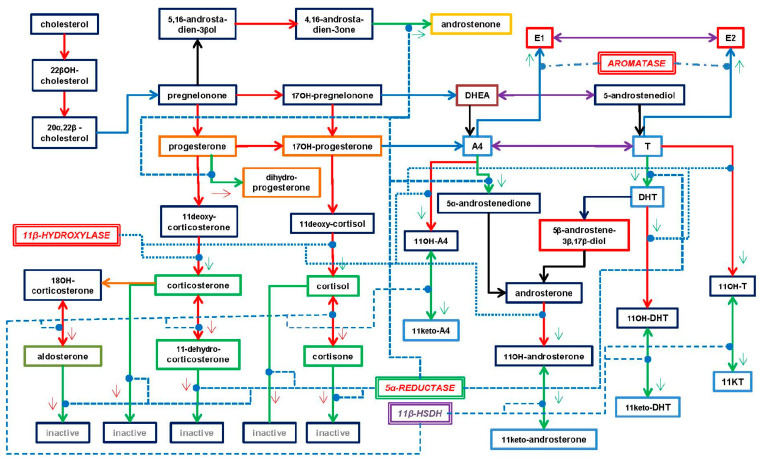
Main shared enzyme activities in the control of glucocorticoid and androgen metabolism in the adrenal gland. Critical role of four key enzymes: aromatase, 5α-reductase, 11β-hydroxylase and 11β-hydroxysteroid dehydrogenase, in the activity and regulation of the main synthesis interrelationships between glucocorticoids, androgens and estrogens. Main glucocorticoids: green-rimmed rectangles (aldosterone: olive green); androgens: dark blue (DHEA: brown) and estrogens: red. The reductive reactions are green arrows, oxidative reactions; red arrows; coenzyme-dependent reversible oxidation-reduction reactions; violet double-pointed arrows (i.e., oxidative or reductive, depending on the tissue redox status). The enzymes are oxidative in red capital letters, reductive in green and equilibrium-oxidation-reductive in violet. Only the four selected key enzymes are listed; those intervening in androgen metabolism have been already described in detail in Figure 1. The actions of enzymes are separately described. The reactions catalyzed have been marked with the color corresponding to the effect elicited: red—oxidation, green—reduction; additional small arrows mark the effects on the product of the overall reaction. Enzyme actions are marked by blue discontinuous lines: aromatase is the point-dash line; 5α-reductase is the dashed line; 11β-hydroxylase is the points line; and 11β-HSDH is the thinner dashed line. Mixed reactions (such as that of aromatase) or actions carried out by non-oxidative-reductive enzymes have been left in black. As explained in the text, the formation of KT and similar compounds are oxidative-activating processes for androgens, but oxidative-inhibiting processes for glucocorticoids. In a reverse way, the action of 5α-reductase enhances the synthesis of DHT and androstenone, but inactivates the oxidized forms of glucocorticoids. Finally, aromatase irreversibly converts A4 or T into estrogens, thus, leaving a narrow (and critically controlled) path for the production of estrogens. In fact, a few key enzymes and a varying metabolic oxidative or reductive ambiance may deeply affect the outcome of the main classes of steroid hormones in a coordinate and partly auto-regulating mechanism.

**Figure 5 ijms-23-11952-f005:**
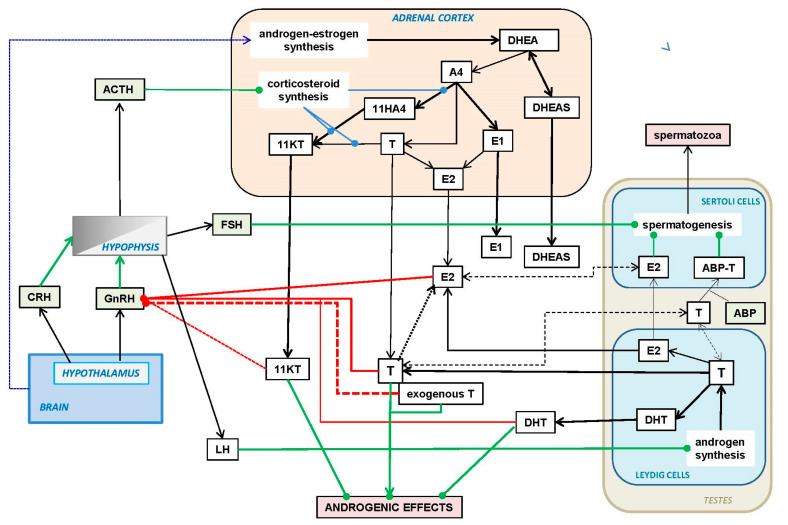
Regulation of the androgen (and estrogen) synthesis by the adrenal cortex and testes. Abbreviations: GnRH: gonadotropin-releasing hormone; CRH: corticotropin-releasing hormone; ACTH: corticotropin; (gonadotropins) FSH: follicle-stimulating hormone; LH: luteotropic hormone; E1: estrone; E2: 3,17β-estradiol; ABP: androgen-binding protein. The other abbreviations have been described in Figure 2. The solid black arrows show paths or relationships. The dotted black line depicts the brain modulation of steroid hormone synthesis by the brain through other additional means (i.e., nervous signals and non-steroidal hormones). The dashed black lines indicate the possibility of direct interchange of pools between tissue and blood. The green lines show stimulatory/activating effects, and the red lines inhibitory/deactivating effects. The effects of exogenous T (i.e., administered as a drug) on GnRH functions overall have been presented as a red dashed line to somehow differentiate it from the tissue-released T.

**Table 1 ijms-23-11952-t001:** Comparison of the physiologic and metabolic effects elicited by the main groups of androgens.

	DHEA	T	AcT	KT	DHT	A4	AP
Gender influences effects	M~F	M > F	~	M~F	M > F	M~F	M > (F)
Binds to AR	(+)	++	~	++	+++	+	(−)
Binds to ER	+	−	−	−	−	−	−
Binds to SHBG	(+) ^1^	+	−	(−)	++	−	~
Male secondary sex effects	−	(+)	~	−	+++	−	−
Aromatase substrate in vivo	−	+	−	+	−	+	−
Activates sexual development	+	+	(+)	+	+	+	−
Increases libido	+ ^F>M^	+^.M&F^	~	−	+ ^M^	~	~
Enhances muscle mass	−	+	(+)	+	+	−	−
Low levels are obesogenic	+ ^M&F^	+ ^M^	(+) ^M^	~	~	+ ^M&F^	~
Lowers insulin resistance	+	+	(+)	−	+	− ^2^	−
Anti-GC effects	++	− ^3^	~	(+) ^4^	~	~	~
Pheromone effects	−	−	−	−	−	−	+

Headings: DHEA (dehydroepiandrosterone); T (testosterone); AcT (17-acyl-testosterone esters); KT (11-keto-testosterone); DHT (dihydrotestosterone); A4 (4-androstenedione); AP (androgenic pheromones). The EA (estrogenic androgens), Ane (5-androsterone) and other androgen metabolites—not described in Section 3—have not been included in the table because of their unclear functions and few (and/or) chemically variable molecule representatives. The distinguishing feature for EA, in any case, is their ability to bind to ER *and* not to AR. Symbols: F = female, M = male (superscript symbols carry the same meaning); ~ = unknown, i.e., no data found/available; + = induces the effect described; − = does not induce the effect described. A symbol between parentheses represents a supposed/hypothetical effect, or an unproven deduction not supported by sufficient (or specific) hard data. Other acronyms; AR (androgen receptor); ER (estrogen receptor); SHBG (sex-hormone binding globulin); GC (glucocorticoid). Cells shadowed in pale green indicate a coincidence in the effects of the marked androgen groups on the same line (i.e., a probably shared effect). The cells with pale brown background show that the given effect is only induced by the marked androgen type (column). Superscript numbers correspond to the notes listed below: ^1^ This effect corresponds to free DHEA, since DHEAS does not bind SHBG [135]. ^2^ Insulin resistance is increased in women with PCOS [136,137]. ^3^ The reverse is true—glucocorticoids tend to antagonize the synthesis and effects of T. ^4^ The synthesis of 11-keto-androgens from 11OH precursors requires oxidation, whereas the formation of active 11OH-corticosteroids requires reduction (Section 5.2); thus, the synthesis of active functional GC is probably not compatible with that of (also active) 11-keto-androgens.

## Data Availability

Not applicable.

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
