# Peer review of "The Roles of Androgens in Humans: Biology, Metabolic Regulation and Health"

_ijms, 2022, doi:10.3390/ijms231911952_

Round 1

Reviewer 1 Report

The review article by Prof Marià Alemany titled “The roles of androgens in humans: Biology, metabolic regulation and health” is a wonderful review article on the structure, biology, and metabolism of androgens in human. I am sure this manuscript will be found extremely useful by many interested readers. In general, I have no concern about the publication of this article, and I would like to endorse its publication. However, there are a few minor issues which need to be fixed before it is published. The following are my recommendations/suggestions to the author:

·       Given the vast number of abbreviations in the manuscript, and the size of the manuscript, I suggest that the author should create a list of abbreviations and insert it to the beginning of the manuscript (after “keywords” and before “introduction”). This will make the readers work easier to locate the full names.

·       On page 9, Figure 3 caption, please write the full names for AF1 and AF2

·       On page 10, Table 1, in the table’s caption, the author should write full names for the abbreviations used in the table.

·       It appears to me that Figure 4 (page 18) for some technical reason is the mirror image of what it should be the actual figure. This issue needs to be fixed.

·       In table 2 (page 28), in the top row of the table, the arrows are directed towards left or right. I assume they are meant to be “up” or “down” arrows. This issue needs to be fixed.

Finally, I would like to congratulate the author for creating such a comprehensive work.

Author Response

Please, read the attached .pdf file, it contains the responses to Reviewers #1 and #2. 

Reviewer 2 Report

The article presents a very important topic, and the content is well explored and important. However, I think that some items it is too extensive, which is clearly visible in the excessive bibliographical references in the article (739).

Figure 4 should be reformulated, it is not understandable, it seems to be inverted.

Another important issue is the non-genomic or rapid effects of testosterone, which are perhaps more important than the classic or genomic effects.

The human pharmacology part is the most important and interesting part of the article since there is nothing explored in this sense. So I think the rest of the article should be summarised and condensed.

Author Response

The responses to both Reviewers, #1 and #2 are included in the same attached .pdf document. Please, read it. Thanks.

Round 2

Reviewer 2 Report

The article has been improved, especially regarding the English language which is now clearer. However, although I accept the author's explanations, I think the article is too long and has too many references. Thus, I think it would be important for the author to cite only the reviews she used and not the 740 articles.

Author Response

Please, read the attached Word file
